# DNA methylation governs the sensitivity of repeats to restriction by the HUSH-MORC2 corepressor

Ninoslav Pandiloski[1,2,3], Vivien Horváth[2,3], Ofelia Karlsson[2,3], Symela Koutounidou[1,3], Fereshteh Dorazehi[1,3], Georgia Christoforidou[1,3], Jon Matas-Fuentes[1], Patricia Gerdes[2,3], Raquel Garza[2,3], Marie E. Jönsson[3], Anita Adami[2,3], Diahann A. M. Atacho[2,3], Jenny G. Johansson[2], Elisabet Englund[4], Zaal Kokaia[3,5], Johan Jakobsson[2,3] & Christopher H. Douse[1,3] ✉

The human silencing hub (HUSH) complex binds to transcripts of LINE-1 retrotransposons (L1s) and other genomic repeats, recruiting MORC2 and other effectors to remodel chromatin. How HUSH and MORC2 operate alongside DNA methylation, a central epigenetic regulator of repeat transcription, remains largely unknown. Here we interrogate this relationship in human neural progenitor cells (hNPCs), a somatic model of brain development that tolerates removal of DNA methyltransferase DNMT1. Upon loss of MORC2 or HUSH subunit TASOR in hNPCs, L1s remain silenced by robust promoter methylation. However, genome demethylation and activation of evolutionarily-young L1s attracts MORC2 binding, and simultaneous depletion of DNMT1 and MORC2 causes massive accumulation of L1 transcripts. We identify the same mechanistic hierarchy at pericentromeric α-satellites and clustered protocadherin genes, repetitive elements important for chromosome structure and neurodevelopment respectively. Our data delineate the epigenetic control of repeats in somatic cells, with implications for understanding the vital functions of HUSH-MORC2 in hypomethylated contexts throughout human development.

Genetic elements may be repetitive either in the sense that there are many copies of the DNA sequence dispersed in the genome, the nucleotide sequence is itself repetitive, or both. Repeats are dynamic stretches of DNA, some of which can mobilize or duplicate. Taken together, repeats comprise more than half of the human genome[1]. The most abundant are interspersed mobile elements called LINE-1 retrotransposons (L1s) that account for 17% of human DNA, a small subset of which continue to generate polymorphisms in the population via retrotransposition events[2,3]. Non-coding, low-complexity repeats such as alpha-satellites (ALRs), which occupy up to 5% of the genome, are not mobile but like L1s are hotspots of recombination and drive genomic complexity within and between individuals[3–5].

Aside from being sources of genetic variation, the functional importance of repeats is increasingly recognized as technologies to

[1]Laboratory of Epigenetics and Chromatin Dynamics, Department of Experimental Medical Science, Wallenberg Neuroscience Center, BMC B11, Lund University, Lund, Sweden. [2]Laboratory of Molecular Neurogenetics, Department of Experimental Medical Science, Wallenberg Neuroscience Center, BMC A11, Lund University, Lund, Sweden. [3]Lund Stem Cell Center, Lund University, Lund, Sweden. [4]Division of Pathology, Department of Clinical Sciences, Lund University, Lund, Sweden. [5]Laboratory of Stem Cells and Restorative Neurology, Department of Clinical Sciences, BMC B10, Lund University, Lund, Sweden. ✉e-mail: christopher.douse@med.lu.se

manipulate and resolve their sequences improve. For example, even though the vast majority are retrotransposition-incompetent[3], L1s impact transcriptional programs by acting as alternative promoters and via changes in chromatin structure[6] – thereby rewiring gene expression[7] and contributing to functional diversification in tissues including the human brain[8]. ALRs form higher-order structures at centromeres that are critical to the structural integrity of chromosomes, while other tandem repeats have widespread effects on gene transcription[9,10]. It is also notable that certain protein and RNA genes are themselves arrayed in clusters, where the genetic structure may be functionally significant[11]. For example, the repetitive array of exons in clustered protocadherin genes are expressed combinatorially, generating a barcoding system that underpins neuronal individuality and complexity in the nervous system[12].

The activity of repeats and repetitive genes is tightly regulated since aberrant control of these elements is associated with genomic instability and disease[13–16]. Packaging of repeats into particular chromatin states is a conserved way to control their transcription and replication. In somatic human cells, repeats are decorated by patterns of DNA and histone methylation that are established early in development[17,18]. Promoter CpG methylation, maintained in somatic cells by DNA methyltransferase 1 (DNMT1) – and trimethylation of lysine 9 in histone H3 (H3K9me3), maintained by histone methyltransferases such as SETDB1, are enriched over repeats and correlate with heterochromatin formation and transcriptional repression[19,20].

The human silencing hub (HUSH) complex has emerged as an important epigenetic regulator of repeats at H3K9me3-marked sites[7,21–24]. Composed of TASOR, MPP8, and Periphilin, HUSH recruits the nuclear ATPase MORC2 and H3K9me3 writer SETDB1 to remodel chromatin[24,25]. Recent data have shown that HUSH also participates as an adapter for co-transcriptional RNA processing complexes, leading to the destruction or termination of targeted transcripts[26–28]. Endogenous genomic repeats targeted by the HUSH-MORC2 co-repressor – that is, L1s and certain repetitive genes[7,21,25,29] – appear to be unified by the presence of long, intronless transcriptional units[30]. Experiments with transgene reporters have shown that transcription is required for recruitment of the complex to the reporter[7,27,30]. A transcription-dependent mechanism of H3K9me3 deposition is reminiscent of evolutionarily ancient systems such as the yeast RNA-induced transcriptional silencing (RITS) complex, and indeed analysis of HUSH protein architectures revealed striking similarities with RITS[21]. Nonetheless, how the HUSH-MORC2 axis interacts with DNA methylation, the central epigenetic regulator of endogenous repeat transcription in somatic mammalian cells, is poorly understood. In mouse embryonic stem cells, where global DNA methylation level depends on media composition[31], retrotransposon repression by HUSH was more pronounced in hypomethylated conditions[32,33] and DNA methylation over a retrotransposon reporter was disrupted by HUSH knockdown[33]. In a human context, key mechanistic studies of HUSH-MORC2 have mainly used transgene reporters and/or cancer cell lines characterized by aberrant DNA methylation. Furthermore, since deletion of DNA methyltransferase 1 (DNMT1), the enzyme that maintains CpG methylation during cell division, is lethal in cancer cells and most somatic mammalian cells[34–36], experiments that provide mechanistic insights into the relationship between HUSH-MORC2 and DNA methylation in repeat regulation are challenging to design.

Here we test the relationship between HUSH-MORC2 and DNA methylation in embryo-derived human neural progenitor cells (hNPCs), a somatic model of brain development with robust levels of DNA methylation that nonetheless tolerates acute removal of DNMT1 and global genome demethylation[37]. Loss of MORC2 or defining HUSH subunit TASOR in hNPCs does not cause widespread misexpression of L1s, all but a handful of which are maintained in a transcriptionally silent state by promoter DNA methylation. However, upon removal of DNA methylation and transcriptional activation, these elements are bound by MORC2 and subject to HUSH-MORC2 restriction such that simultaneous depletion of DNMT1 and MORC2 leads to massive accumulation of LINE-1 transcripts. We provide evidence of the same mechanistic hierarchy at pericentromeric ALRs and clustered protocadherin genes, repetitive elements important for chromosome stability and nervous system development respectively. Together our data delineate the epigenetic control of repeats in somatic cells, with implications for understanding the vital functions of the HUSH-MORC2 axis in hypomethylated contexts throughout human development.

## Results

### Epigenomic profiling of the human neural progenitor cell model

To investigate the relationship of the HUSH-MORC2 axis with DNA methylation in epigenetic repeat regulation (Fig. 1A) we used a somatic human neural progenitor cell (hNPC) line derived from fetal brain 6 weeks post-conception[38], which represents a timepoint of post-implantation embryonic development when DNA methylation has been fully re-established[17,37]. These cells have the characteristics of neuroepithelial-like stem cells[38]. We confirmed expression of the NPC markers Nestin and SOX2, and the neural differentiation capacity of the cells (Fig. 1B and Supplementary Fig. 1A, B). Analysis of the hNPC proteome detected all core HUSH complex components (TASOR, MPP8, PPHLN1), its effectors (MORC2, SETDB1), and maintenance DNA methyltransferase DNMT1 (Supplementary Fig. 1C)[37]. Single nuclei RNA-seq analysis[8] of six fetal forebrain tissue samples (7.5–10.5 weeks post-conception) illustrated widespread expression of the genes encoding these factors in all identified cell types of the developing human brain (Supplementary Fig. 1D).

Given our focus on repetitive elements in this study, we used long-read Oxford Nanopore (ONT) sequencing to resolve DNA methylation patterns in the hNPC genome[39]. We found that more than 80% of CpGs are methylated genome-wide, including over L1, SVA, and HERV-K retrotransposon families (Fig. 1C), typical of somatic cells[40]. To investigate the histone methylation status of the hNPC model and compare it to actual human tissue, we performed CUT&RUN profiling of H3K4me3 (enriched at active transcription start sites) and heterochromatin mark H3K9me3 (enriched over repeats) in hNPCs and human fetal forebrain samples ($n$ =2, 7- and 10-weeks post conception) alongside non-targeting IgG controls (Fig. 1D). More than 90% of H3K4me3 and H3K9me3 peaks called in fetal tissue samples were also occupied by H3K4me3 and H3K9me3 in the hNPC model, and vice versa (Fig. 1E, F and Supplementary Fig. 1E, F). The hNPCs cannot fully describe the epigenetic status or heterogeneity in the developing human brain – and are isolated from a timepoint that slightly precedes the tissue samples analyzed by snRNA-seq and CUT&RUN. Nonetheless, these experiments suggest that hNPCs represent a useful model for querying epigenetic mechanisms of repeat regulation in human brain development and confirm that the DNA methylation patterns of somatic human cells are present in the hNPC model.

### Targeted depletion of HUSH and MORC2 with CRISPRi

To inhibit expression of MORC2 or defining[21] HUSH subunit TASOR we used a bulk lentiviral CRISPR interference (CRISPRi) approach. We co-expressed a dCas9-KRAB-GFP fusion protein with gRNAs targeted to the relevant TSS, or a non-targeting control (Fig. 1G). Underpinning CRISPRi is chromatin remodeling of the targeted TSS through KRAB repressor binding, accumulation of promoter H3K9me3 and transcriptional silencing. Following 10 days' expansion, epigenome profiling in the CRISPRi lines indeed revealed targeted accumulation of H3K9me3 and concomitant loss of H3K4me3 over the relevant TSS (Fig. 1H). This corresponded to essentially complete depletion of MORC2 and TASOR transcript (Fig. 1I) and protein (Fig. 1J) levels. We note that the *MORC2* locus is itself HUSH-repressed[21], and indeed saw

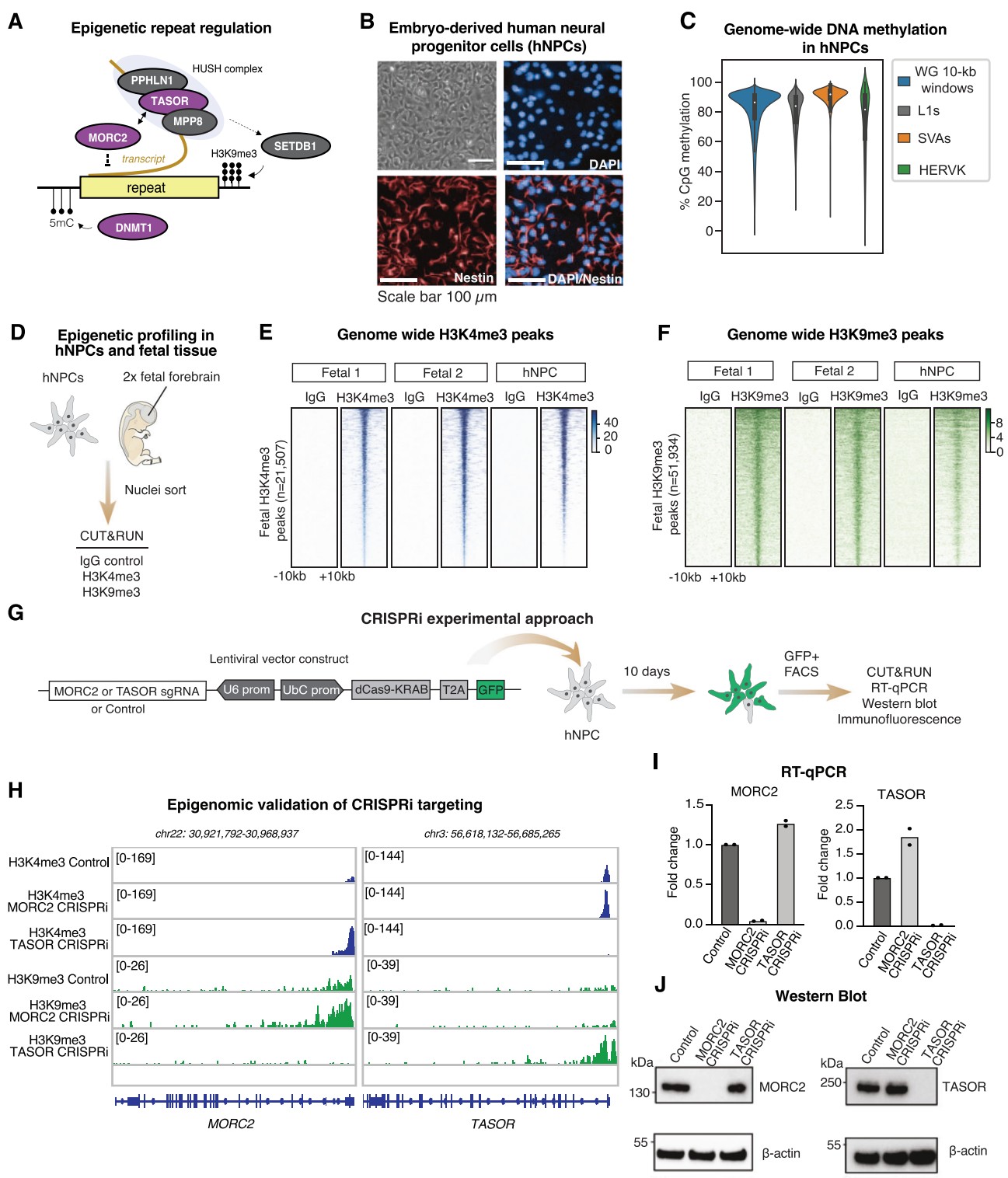

clear evidence for loss of H3K9me3 over the *MORC2* promoter in TASOR-CRISPRi hNPCs, correlated with an increase in transcript levels, confirming a functional knockdown. Neither MORC2 nor TASOR depletion caused an obvious morphological or fitness change in hNPCs and marker gene expression was consistent across treatment and control groups, suggesting that cell identity was not perturbed by loss of these factors (Supplementary Fig. 1G, H). Thus, CRISPRi proved a successful strategy for targeted and acute depletion of HUSH and MORC2 in hNPCs.

## DNA methylation, but not HUSH-MORC2, controls young L1 transcription in hNPCs

We began by assessing the relationship between HUSH-MORC2 and DNA methylation in the regulation of L1 retrotransposons. Full-length L1s (FL-L1s, here defined as > 6 kb) contain a promoter in their 5′ UTR that drives production of a bicistronic mRNA encoding two open reading frames (ORF1p and ORF2p, Fig. 2A). Since L1 integrants accumulate mutations over evolutionary time, elements can be classified according to evolutionary age, including human-specific (L1Hs) and

**Fig. 1 | Studying the epigenetic control of repeats in a somatic cell model of brain development. A** Simplified model of epigenetic transcriptional regulation of genomic repeats by DNA methylation and the HUSH-MORC2 corepressor. Factors shaded in purple are subject of particular focus in this study. **B** Brightfield image and immunostaining of the hNPC line used in this study illustrating Nestin expression. Imaging was repeated frequently during the study to confirm cell morphology and marker expression. **C** Assessment of genome-wide CpG methylation according to whole-genome Oxford Nanopore sequencing of the hNPC line ($n = 1$) at approximately ×40 coverage. The violin plots represent the median CpG methylation of the given set of genomic intervals and span the interquartile range. **D** Schematic of CUT&RUN epigenome profiling experiments in human fetal forebrain tissue ($n = 2$) and hNPCs ($n = 3$). **E, F** Heatmaps illustrating CUT&RUN signal enrichment of H3K4me3, H3K9me3 and a non-targeting IgG control in fetal

forebrain samples and hNPCs plotted over consensus peaks called in fetal samples. Displayed are the genomic regions spanning ±10-kb from the peak center. **G** Schematic of CRISPRi experiments to deplete MORC2 or HUSH subunit TASOR. **H** Genome browser snapshots of *MORC2* (left) and *TASOR* (right) genes illustrating epigenome editing by the CRISPRi technology, as measured by CUT&RUN sequencing (H3K4me3, H3K9me3) relative to non-targeting control. **I, J** Reverse transcription quantitative PCR (RT-qPCR) and Western blot analysis of MORC2 (left) and TASOR (right) transcript and protein levels in MORC2 and TASOR CRISPRi cells, relative to a non-targeting control, 10 days post-transduction. Experiments were repeated at least once with similar results. Data points shown for the RT-qPCR are two independent replicates, each constituting the average of three technical replicates. Source data are provided in a Source Data file.

hominoid-specific (L1PA2-L1PA4) subfamilies. FL-L1s from these youngest subfamilies retain the capacity to be actively transcribed (for example in pluripotent cells), but are silenced by promoter DNA methylation in hNPCs[37]. Clones of K562 cells that lacked components of HUSH or MORC2 previously showed pronounced upregulation of these young L1 subfamilies and production of L1-derived protein[7]. However, K562 cells are essentially completely demethylated[41]. To investigate transcriptional regulation of L1s by HUSH-MORC2 in hNPCs, where young FL-L1s are robustly methylated – and remain so upon MORC2 depletion (Fig. 2B and Supplementary Fig 2) – we used a 2 × 150 bp, polyA-enriched stranded library preparation for bulk RNA-seq following MORC2 or TASOR CRISPRi, with a reduced fragmentation step to optimize library insert size for repeat analysis. Using this approach reads can be mapped uniquely to most individual L1 instances, except for some integrants belonging to the L1Hs subfamily and polymorphic alleles that are not annotated in the reference assembly[8,37]. In differential expression analysis of L1s and other repeats we routinely used two approaches – 'unique mapping', where ambiguously mapped reads were discarded, and 'multi-mapping' where those reads were retained and assigned to subfamilies by the TEtranscripts software[42]. To remove DNA methylation we took advantage of a validated CRISPR-Cas9 vector to disrupt the catalytic domain of DNMT1, which is known to activate evolutionarily young FL-L1s in these cells (Fig. 2C)[37]. Given the mappability challenges for the youngest L1s, we further complemented the short-read RNA-seq analysis with long-read Nanopore cDNA-seq (including a strand-switching step for 5′ enrichment), and H3K4me3 CUT&RUN analysis to provide orthogonal validation of expression patterns observed. The latter experiment is particularly useful since the H3K4me3 signal spreads from the L1 promoter to the upstream genome, which aids unique mapping.

Strikingly, we did not observe widespread transcriptional deregulation of L1s in hNPCs lacking MORC2 or TASOR: we detected only modest transcriptional changes amongst the youngest L1 subfamilies (Fig. 2D), no accumulation of H3K4me3 over the 5′ UTR promoter of young FL-L1s (Fig. 2E), nor depletion of H3K9me3 (Fig. 2F). We also could not detect ORF1p expression by Western blotting (Fig. 2G). By contrast, following disruption of DNMT1, pronounced transcriptional activation of FL-L1s was accompanied by H3K4me3 over young FL-L1 promoters and ORF1p expression (Fig. 2D, E, G). These expression patterns were supported by long-read cDNA sequencing reads accumulating over the youngest L1 instances only in the DNMT1-KO condition (Fig. 2H,I). H3K9me3, which is present alongside CpG methylation over young FL-L1s in control hNPCs, was also not significantly affected in the DNMT1-KO cells over the 10-day timeframe of the experiment (Fig. 2F), illustrating that the presence of H3K9me3 is compatible with expression of these elements. Finally, we looked at interferon-stimulated genes, since L1 derepression by HUSH has been associated with stimulation of the innate immune system[43]. Correlated with the lack of L1 deregulation, we did not observe such stimulation in MORC2 or TASOR CRISPRi hNPCs, although expression of *IRF1*, *IFIH1*

(encoding MDA5), and *DDX58* (encoding RIGI) were upregulated in DNMT1-KO cells (Fig. 2J).

## HUSH-MORC2-sensitive L1s are part of genic transcripts or partially demethylated

Our data so far illustrate that in hNPCs, evolutionarily young L1 families are transcriptionally silenced by promoter DNA methylation, rendering them mostly insensitive to HUSH-MORC2 binding and/or H3K9me3 deposition. At the developmental timepoint captured by this somatic model, H3K9me3 maintenance over silent (DNA-methylated) young FL-L1s thus does not appear to depend on HUSH-MORC2. One exception to this trend are L1s whose transcription is driven by upstream promoters. Indeed, when we assessed global changes in H3K9me3, we observed loss of H3K9me3 over hundreds of sites upon MORC2 or TASOR CRISPRi, mostly older L1PA elements in introns of transcribed genes (Supplementary Fig. 3A−E). Notably, 88% of these occupied the same strand as the host gene, a striking observation given that intronic L1s predominantly lie in the opposite orientation to their host genes (outweighing sense insertions by about 2:1)[44] (Supplementary Fig. 3F). The selectivity for sense insertions in transcribed genes supports the model that transcription of the A-rich L1 forward strand is a determinant of HUSH recognition and H3K9me3 deposition[30], regardless of whether transcription is element-derived or (as in these examples) driven from an upstream gene promoter. CUT&RUN using an optimized crosslinking protocol enabled detection of direct MORC2 binding at around half (262/532) of targeted sites using conservative cutoffs (Supplementary Fig. 3G, H). Genes harboring HUSH-MORC2 bound FL-L1s were modestly upregulated relative to those containing unbound FL-L1s (Supplementary Fig. 3I).

We next investigated other L1 instances that did show sensitivity to loss of HUSH-MORC2 in hNPCs. In accordance with subfamily-wide results presented above, when unique mapping was enforced in transcriptomic analysis, we found that only 19 FL-L1s were significantly upregulated upon MORC2 or TASOR CRISPRi (Supplementary Fig. 4A). Analysis of the sequences and epigenetic status of these elements showed that only three had both an intact YY1 binding site (required for accurate transcription initiation) and evidence of H3K4me3 enrichment over their promoter, suggesting that the remaining elements were transcribed from other promoters lying upstream of the L1. Interestingly, upon examination of the promoters of these three candidate elements in our Nanopore sequencing dataset, we could detect a proportion of demethylated reads in all cases (Fig. 3A), suggesting that the elements are incompletely methylated in hNPCs. We therefore hypothesized that DNA methylation status, by controlling L1 promoter activity, determines the sensitivity of FL-L1s to HUSH-MORC2 restriction. To test this, we first compared transcriptomic changes in three published datasets of TASOR depletions in cancer cells with drastically different DNA methylation levels over L1s. In HeLa cells, which (despite global chromosomal instability) have broadly comparable L1 methylation levels to what we observe in hNPCs,

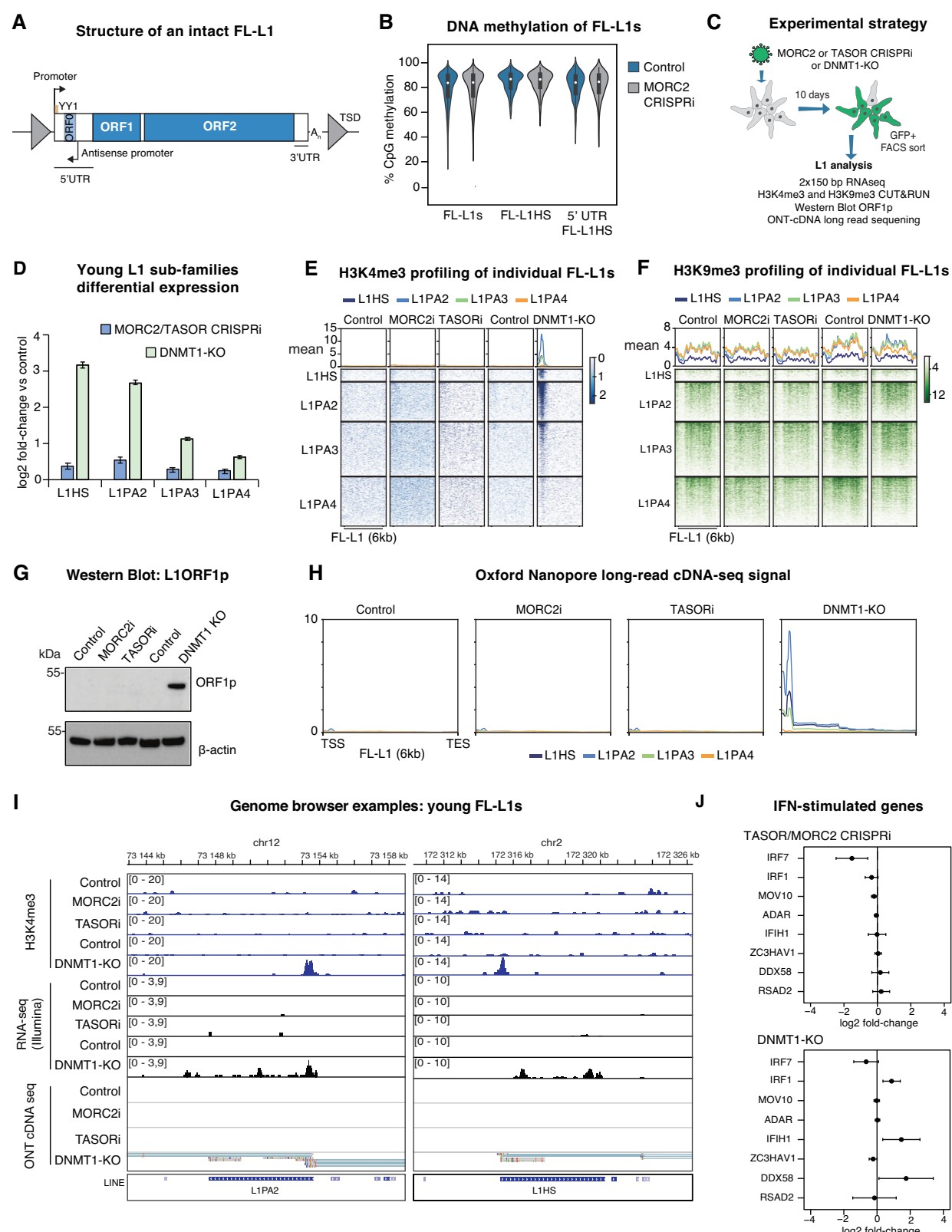

**A** Structure of an intact FL-L1

**B** DNA methylation of FL-L1s

**C** Experimental strategy

**D** Young L1 sub-families differential expression

**E** H3K4me3 profiling of individual FL-L1s

**F** H3K9me3 profiling of individual FL-L1s

**G** Western Blot: L1ORF1p

**H** Oxford Nanopore long-read cDNA-seq signal

**I** Genome browser examples: young FL-L1s

**J** IFN-stimulated genes

modest L1 deregulation was likewise observed[28,43]. In K562 cells, which are hypomethylated[41] and where L1s are transcribed at steady-state, L1 upregulation was substantially more pronounced (Supplementary Fig. 4B)[7]. In K562 cells this is accompanied by MORC2 binding to young FL-L1s, as assessed by reanalysis of published ChIP-seq data[7] (Supplementary Fig. 4C). To test the model directly in hNPCs, we next sought to measure MORC2 binding to L1s upon genome demethylation

(Fig. 3B). We used CRISPRi as an alternative approach to DNMT1 knockout reported previously, validating two gRNAs for successful inhibition of *DNMT1* transcription by RT-qPCR (Supplementary Fig 5A). Global loss of DNA methylation in DNMT1 CRISPRi cells was confirmed by 5mC immunostaining (Supplementary Fig. 5B). RNA-seq analysis showed pronounced derepression of the evolutionarily youngest L1 subfamilies and imprinted or germline-restricted genes, and mild

**Fig. 2 | Young FL-L1s are silenced by DNA methylation but not HUSH-MORC2 in hNPCs. A** Genetic structure of a full-length (>6-kb) intact L1 retrotransposon. TSD, target site duplication; UTR, untranslated region; ORF, open reading frame. **B** Nanopore DNA methylation analysis over hg38 reference FL-L1s in control (*n* = 1) and MORC2 CRISPRi hNPCs (*n* = 1). The violin plots represent the median CpG methylation of the given set of genomic intervals and span the interquartile range. **C** Schematic of L1 analysis of MORC2 and TASOR CRISPRi and DNMT1-KO hNPCs. **D** Log₂-fold-change (LFC) of young L1 subfamilies measured by RNA-seq in MORC2 or TASOR CRISPRi hNPCs (*n* = 7) versus controls (*n* = 4) using the TEtranscripts software. DNMT1-KO RNA-seq data in hNPCs (*n* = 3 in control group; *n* = 3 in treatment group) taken from Jönsson et al.[37]. In each case the error bars represent +/- LFC standard error calculated by DESeq2 taking all samples into account. **E, F** Heatmaps illustrating CUT&RUN signal enrichment of H3K4me3 and H3K9me3 in control, MORC2 CRISPRi, TASOR CRISPRi and DNMT1-KO hNPCs, plotted over young full-length L1PA families sorted by evolutionary age (top to bottom). Only

signal with mapQ score >10 was used to generate signal matrices. In all cases experiments were performed 10 days post-transduction and the experiments repeated at least once with similar results. **G** Western blotting for L1 orf1p expression in control, MORC2 CRISPRi, TASOR CRISPRi and DNMT1-KO hNPCs. The experiment was repeated once with similar results. **H** Mean signal of Nanopore long-read cDNA sequencing in control (*n* = 1), MORC2 CRISPRi (*n* = 1), TASOR CRISPRi (*n* = 1) and DNMT1-KO (*n* = 1) hNPCs aligning to young full-length L1s. Only uniquely-mapping reads were retained. **I** Two genome browser examples of FL-L1s upregulated by loss of DNMT1 but not MORC2 or TASOR. In all tracks only uniquely-mapping reads were retained. **J** Differential expression analysis of selected interferon (IFN) stimulated genes in the TASOR and MORC2 CRISPRi treatments (*n* = 7) versus control hNPCs (*n* = 4) and in DNMT1-KO hNPCs (*n* = 3) versus control (*n* = 3). Shown are LFC values ±LFC standard error calculated by DESeq2, taking all samples into account. The DNMT1-KO RNA-seq data was taken from Jönsson et al.[37]. Source data are provided in a Source Data file.

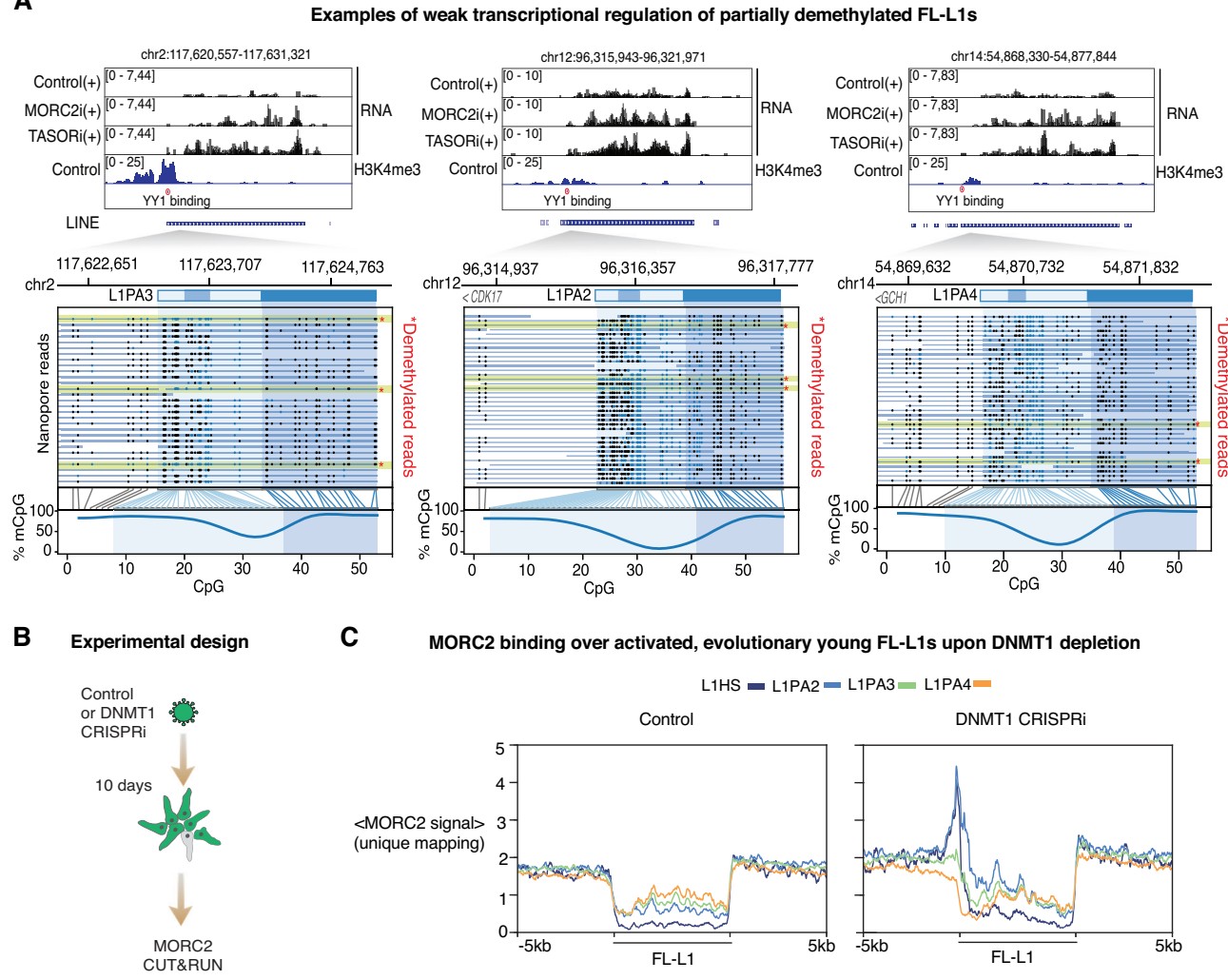

**Fig. 3 | DNA methylation controls the activity of HUSH-MORC2 at FL-L1s.**
**A** Three examples of FL-L1s weakly upregulated by MORC2 and TASOR CRISPRi showing promoter H3K4me3 in control hNPCs and an intact YY1 binding site (upper). Nanopore sequencing reads suggest incomplete DNA methylation over the promoter in control hNPCs (lower). In the Nanopore reads CpGs in filled black circles are called as methylated; open blue circles are unmethylated. **B** Measurement of MORC2 L1 binding upon genome demethylation by DNMT1

CRISPRi. **C** Summary plots illustrating CUT&RUN signal enrichment of MORC2 in control and DNMT1 CRISPRi hNPCs, plotted over young full-length L1 subfamilies sorted by evolutionary age. Only uniquely-mapped reads were used to generate signal matrices. Displayed are the genomic regions spanning ±5-kb from the 6-kb L1 element. The dip over the body of the elements is due to lower average unique mappability of these youngest elements relative to the surrounding genome.

induction of interferon-stimulated genes, consistent with DNMT1-KO hNPCs[37], while marker gene expression did not suggest that cell identity was perturbed (Supplementary Fig. 5C–F). In the MORC2 CUT&RUN experiment, we found little evidence of MORC2 binding to the evolutionarily youngest FL-L1s in control hNPCs, consistent with the model that these elements are largely ignored at steady-state when silenced by DNA methylation. However, upon depletion of DNMT1 and concomitant genome demethylation, we could observe clear accumulation of MORC2 over the 5' end of those youngest FL-L1s (Fig. 3C).

## DNA methylation status controls young L1 sensitivity to HUSH-MORC2 restriction

Next, to test the functional impact of MORC2 on the transcription of demethylated L1s, we designed double CRISPRi experiments to deplete DNMT1 in combination with MORC2 (Fig. 4A). We compared the transcriptomes of the MORC2/DNMT1 double CRISPRi treatment to non-targeting controls, and these datasets to individual MORC2 and DNMT1 CRISPRi experiments. Loss of MORC2 and DNMT1 expression was confirmed in the RNA-seq data (Fig. 4B). Using unique mapping to discriminate FL-L1s from truncated copies, we observed upregulation of 2504 elements (LFC >2, padj <0.05) upon simultaneous loss of MORC2 and DNMT1, more than a third of all FL-L1s in the hg38 reference genome, with no obvious preference for genomic context (Supplementary Fig. 6A). The expression of L1s from human- and primate-specific subfamiles (L1HS, L1PA2, L1PA3, L1PA4) was dramatically increased in the double CRISPRi experiment, whether the analysis was done at a subfamily level (Fig. 4C) or at an individual element level (considering only FL-L1s and using unique mapping) (Fig. 4D, E). These observations were validated by CUT&RUN experiments showing increased H3K4me3 levels over young FL-L1 promoters in the double CRISPRi cells (Fig. 4F, G). Importantly, the deregulation of long terminal repeat (LTR) transposon families (e.g. LTR12C, Harlequin) and imprinted or germline-restricted genes (e.g. *XIST, DAZL, MAEL, NNAT*) – activated in cells lacking DNMT1 but not thought to be controlled by HUSH-MORC2 – were unchanged between the DNMT1 and MORC2/DNMT1 CRISPRi treatments (Supplementary Fig. 6B, C). We did, though, observe evidence of compounded effects of the double CRISPRi at several L1-fusion genes[37] i.e. where a young L1 provides an alternative promoter (e.g. *GJB7, GABRR1, WDR72*) (Supplementary Fig. 6D). We also asked whether the same mechanistic hierarchy applied at non-reference polymorphic L1s. To answer this question we leveraged our whole-genome Nanopore sequencing and the Transposons from Long DNA Reads (TLDR) package[39] to annotate polymorphic FL-L1s. We identified 66 L1 insertions longer than 6 kb, of which 53 passed filters such as read coverage and target-site duplication (TSD) sequence. Notably, the majority (40/53) of these insertions were heterozygous, as assessed by reads with gaps spanning the insertion site (Supplementary Fig 7A). We first confirmed that the 5' UTR of homozygous polymorphic FL-L1s were fully covered by CpG methylation (Supplementary Fig. 7B). By plotting the H3K4me3 CUT&RUN data from Control, DNMT1-CRISPRi and MORC2/DNMT1-CRISPRi hNPCs centered on the TSDs in the hg38 reference genome, we could clearly detect the same pattern of activation over most of these 53 polymorphic FL-L1 alleles upon genome demethylation and further derepression upon concomitant MORC2 and DNMT1 loss, based on the spreading of the H3K4me3 signal to the surrounding genome (Supplementary Fig. 7C). Re-mapping the epigenomics datasets to a custom-built genome also enabled us to observe these patterns over the 5'UTR of individual L1 polymorphisms (Supplementary Fig. 7D).

Taken together, these data demonstrate that loss of DNA methylation in hNPCs activates transcription of young FL-L1s, which become targets of MORC2 (Fig. 4G, H). Simultaneous loss of DNMT1 and MORC2 therefore causes a massive accumulation in the levels of young FL-L1 transcripts.

## HUSH-MORC2 and DNA methylation co-regulate ALRs and clustered protocadherins in hNPCs

Having established that DNA demethylation via loss of DNMT1 sensitizes L1 elements to HUSH-MORC2 restriction, we next considered whether this was a general effect at other classes of repetitive element. In subfamily-level analysis of RNA-seq data upon loss of MORC2 or TASOR individually, the alpha-satellite-like repeat (ALR) was by far the most upregulated class (Fig. 5A). We thus considered the DNA methylation status of ALRs. Although ultra-long (>100-kb) sequencing reads are required to uniquely map higher order arrays of ALRs at centromeres[45], the more scattered distribution of ALRs in pericentromeric regions allowed for mapping of thousands of Nanopore reads (N50, 14-kb) that span relatively short ALR arrays and/or are interspersed in unique sequence contexts. We found that CpGs in pericentromeric regions containing ALRs were around 50% methylated, significantly below the whole genome average (Fig. 5B, C). Thus, ALRs are somewhat hypomethylated and subject to HUSH-MORC2 restriction at steady-state in hNPCs. Notably, ALRs were also markedly upregulated in hNPCs lacking DNMT1, illustrating that CpG methylation is involved in controlling ALR transcription (Fig. 5D). DNA methylation status was unchanged over ALRs in cells depleted of MORC2, as was observed at L1s (Supplementary Fig. 8A). We also saw a compounded effect on ALR transcript levels upon simultaneous depletion of MORC2 and DNMT1 relative to either treatment individually (Fig. 5D). Together these results show that: (i) ALRs are hypomethylated and transcribed in hNPCs despite being embedded in a heterochromatin environment and (ii) HUSH-MORC2 controls ALR transcript levels at these regions. Since ALRs are markedly less methylated than the rest of the genome in hNPCs, and DNMT1 depletion further upregulates ALR transcription, our data would predict that robust DNA methylation of pericentromeric ALRs would silence these repeats and render them insensitive to loss of HUSH-MORC2. Whether particular sequence features (e.g. high AT-content) of ALRs underlie the sensitivity of these repeats to HUSH-MORC2, or change the dynamic between CpG methylation and repeat transcription at ALRs, requires further investigation.

Lastly, we considered the effect of HUSH-MORC2 on gene regulation in hNPCs. Of the 24 most upregulated genes in hNPCs lacking MORC2 or TASOR (LFC > 2, Supplementary Fig. 8B), 10 were in (ALR-rich) pericentromeric regions (Supplementary Fig. 8C). Genes within 1Mb of autosomal centromeres were significantly more strongly upregulated than those outside these regions (Supplementary Fig. 8D). Gene ontology analysis of all significantly upregulated genes ($n = 373$) revealed an enrichment of terms such as cell-cell adhesion and synapse organization (Fig. 5E), driven by protocadherin genes from the beta (*PCDHβ*) and gamma (*PCDHγ*) clusters on chromosome 5. This was accompanied by a gain of H3K4me3 over existing transcriptional start sites and the appearance of several new H3K4me3 peaks in both beta and gamma clusters (Fig. 5F). We observed near-total ablation of H3K9me3 over the *PCDHγ* cluster upon loss of MORC2 or TASOR but, notably, no major change over the *PCDHβ* cluster, despite transcriptional and H3K4me3 changes over both clusters (Fig. 5F). MORC2 binding was identified at the locus, dominated by a large peak over *TAF7*, a single exon gene known to be HUSH-targeted[24] that lies between the beta and gamma *PCDH* clusters (Fig. 5F). CUT&RUN data generated from bulk fetal forebrain and neurons from post-mortem adult cortical tissue ($n = 2$) revealed a large (~750-kb) H3K9me3 domain over all three *PCDH* clusters (Supplementary Fig. 9A). While we cannot prove that this domain is HUSH-MORC2-dependent in the human brain in vivo, we note that the complex is robustly expressed in both fetal and adult forebrain tissue according to in-house snRNA-seq data (Supplementary Figs. 1D and 9B)[8,46].

We finally asked whether the same mechanistic hierarchy as observed at ALRs and L1s also applied at clustered *PCDH* genes. Loss of DNMT1 upregulated *PCDH* transcription and promoter H3K4me3

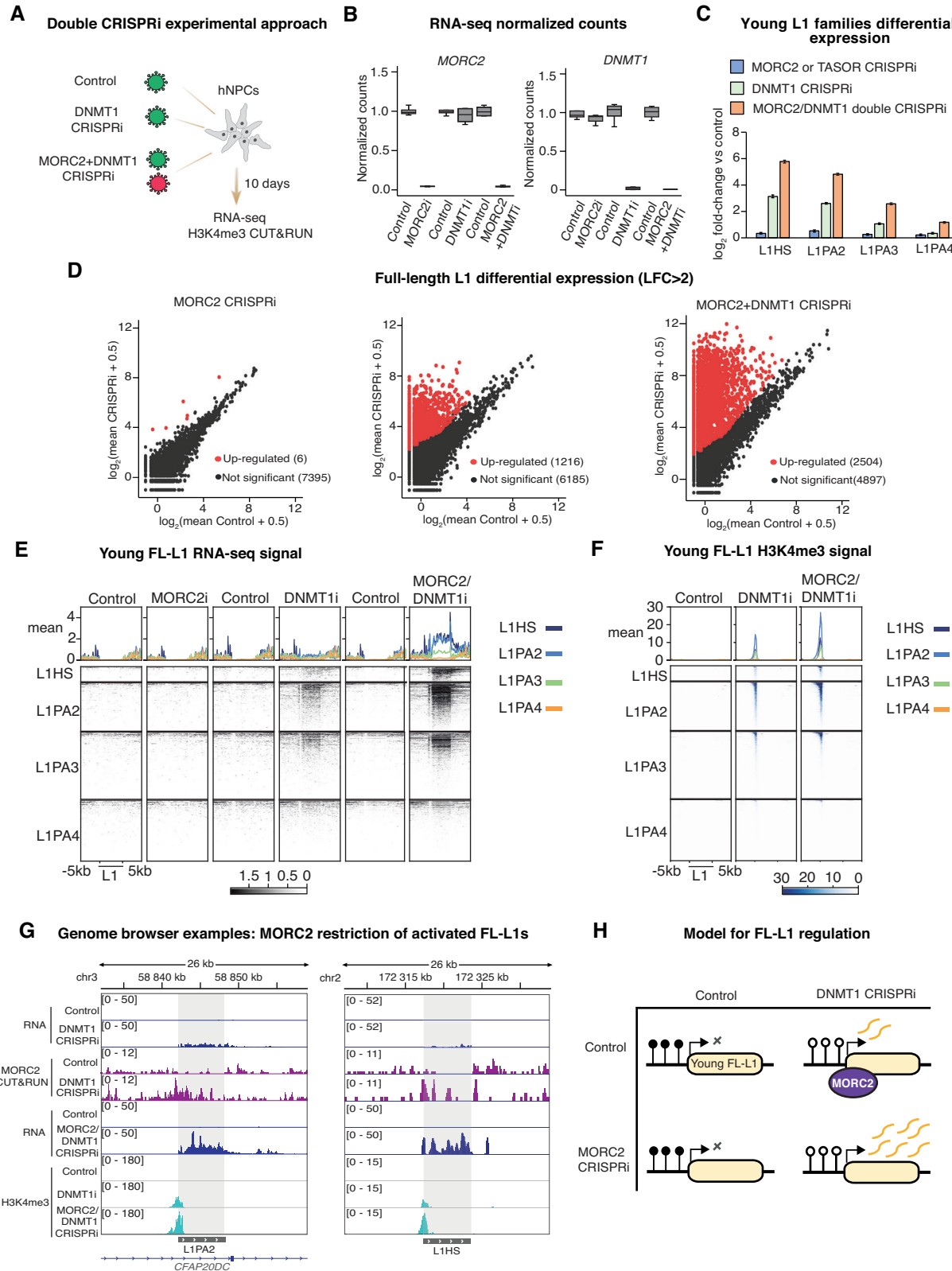

without affecting local H3K9me3 (Supplementary Fig. 9C). MORC2 CRISPRi did not affect DNA methylation over the *PCDH* cluster (Supplementary Fig. 9D). The double knockdown of MORC2 and DNMT1 caused greater transcriptional output from *PCDH*s than either depletion individually (Fig. 5G). Together these data suggest that *PCDH* genes, like L1s and ALRs, are subject to a dual layer of epigenetic control in the developing human nervous system.

## Discussion

As well as defending vertebrate genomes by silencing diverse exogenous genetic elements, the HUSH complex participates in the epigenetic control of endogenous repeats including L1 retrotransposons and repetitive genes[7,21,24,30,47–49]. HUSH recruits the nuclear ATPase MORC2 that, together with H3K9me3-writer SETDB1, remodels chromatin at target loci[7,22,25,28,50]. Recently, a series of studies have shown

**Fig. 4 | Genome demethylation sensitizes young L1s to restriction by MORC2.**
**A** Strategy for simultaneous MORC2 and DNMT1 CRISPRi. **B** Box plots showing normalized read counts of bulk RNA-seq experiments in control ($n = 4$), MORC2 CRISPRi ($n = 3$), control' ($n = 4$), DNMT1 CRISPRi ($n = 4$), control'' ($n = 4$), and MORC2/DNMT1 double CRISPRi ($n = 4$) hNPCs. Central bands denote medians. Boxes represent the interquartile range and whiskers represent maxima and minima. **C** Log$_2$-fold-change (LFC) of young L1 subfamilies measured by RNA-seq in MORC2 or TASOR CRISPRi hNPCs ($n = 7$, blue), DNMT1 CRISPRi ($n = 4$, green), and MORC2/DNMT1 double CRISPRi ($n = 4$, orange) versus controls ($n = 4$ for each independent experiment) using TEtranscripts. In each case the error bars represent

±LFC standard error calculated by DESeq2 taking all samples into account. **D** Mean plots of uniquely-mapped RNA-seq read counts for different treatments versus their respective controls over reference FL-L1s. Red points indicate that the element satisfied both log$_2$-fold-change >2 and padj < 0.05 cutoffs as calculated by DESeq2. **E, F** Heatmaps of RNA-seq and H3K4me3 CUT&RUN signal plotted over young full-length L1PA families sorted by evolutionary age (top to bottom) in the given hNPC treatments. In all cases only uniquely-mapped reads were retained. **G** Examples of young FL-L1 transcriptional and epigenomic changes upon DNMT1 and MORC2/DNMT1 depletion. **H** Schematic model for regulation of young FL-L1s by DNA methylation and MORC2.

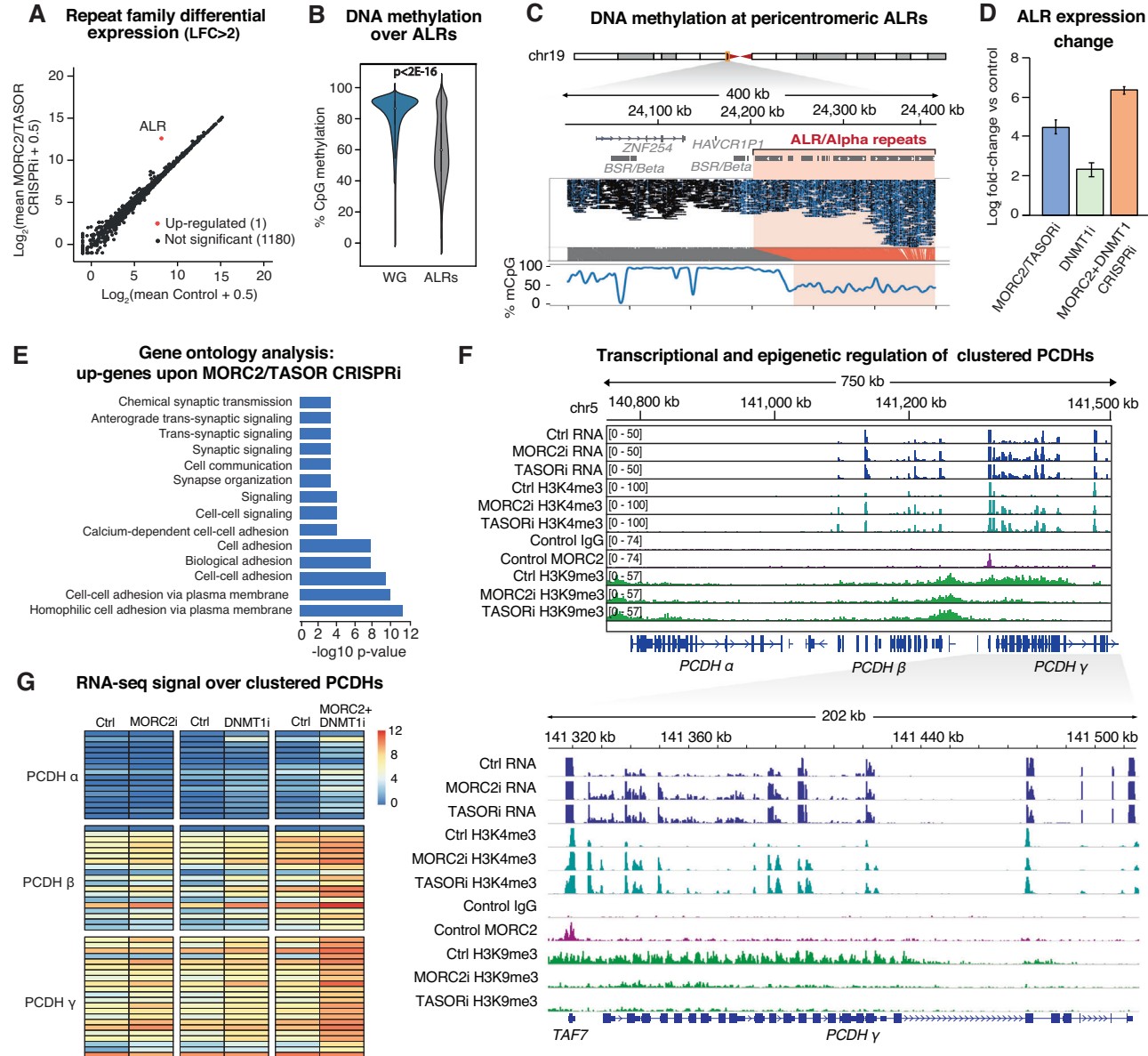

**Fig. 5 | HUSH-MORC2 and DNA methylation control ALRs and protocadherins.**
**A** Mean plots of RNA-seq data in Control ($n = 4$) and MORC2 and TASOR CRISPRi hNPCs ($n = 7$) illustrating differential repeat expression based on TEtranscripts. ALR is the only subfamily with |log$_2$ fold-change| >2 and padj <0.05. **B** Average CpG methylation in reads mapping uniquely to pericentromeric regions relative to the genome average (10-kb bins), illustrating relative ALR hypomethylation in this hNPC line ($n = 1$). The violin plots represent the median CpG methylation of the given set of genomic intervals and span the interquartile range. A two-sided Wilcoxon rank sum test with continuity correction was used to compare the methylation levels between the two sets of regions. **C** Example of DNA methylation status in ALR-rich pericentromeric region on chr19. In the Nanopore reads CpGs in filled

black circles are called as methylated; open blue circles are unmethylated. **D** Log$_2$-fold-change (LFC) of ALRs measured by RNA-seq in MORC2 or TASOR CRISPRi hNPCs ($n = 7$, blue), DNMT1 CRISPRi ($n = 4$, green) and MORC2/DNMT1 double CRISPRi ($n = 4$, orange) versus controls ($n = 4$ for each independent experiment). In each case the error bars represent ±LFC standard error calculated by DESeq2 taking all samples into account. **E** Gene ontology analysis of 373 differentially-expressed genes in MORC2 and TASOR CRISPRi hNPCs (no LFC cutoff, DESeq2 padj < 0.05). **F** Genome browser snapshot of the clustered protocadherin locus on chr5 illustrating transcriptional and chromatin regulation by HUSH-MORC2, with a zoom-in of the *TAF7* gene and *PCDH* gamma cluster. **G** Heatmaps of average log$_2$-transformed counts over clustered *PCDH* genes in the given treatments.

that HUSH impacts the transcriptional output of its targets by binding to nascent RNA and to nuclear complexes responsible for transcript degradation and termination[26–28,30]. However, a significant context to many of these findings is that they been made in cells with hypomethylated genomes. For example, HUSH-MORC2 was originally identified as an L1 regulator in K562 cells[7], where the L1 promoter is essentially completely demethylated[41]. Since DNA methylation is strongly correlated with transcriptional silencing of repeats[17,51], and transcription is a critical determinant of HUSH-MORC2 targeting, whether and how these two pathways cooperate in the context of epigenetic repeat regulation is an important question that has remained largely unanswered.

In this study, we systematically tested the relationship between HUSH-MORC2 and DNA methylation in controlling transcription of endogenous genomic repeats and repetitive genes in the human genome. To do this we took advantage of a somatic cellular model of early brain development where DNA methylation is robustly established but which, surprisingly, remains proliferative upon global genome demethylation by DNMT1 knockout[37]. Through locus-specific analysis integrating Oxford Nanopore DNA methylation with bulk long- and short-read transcriptomic and histone methylation profiling, we identified that a handful of partially-demethylated L1 copies are weakly-transcribed and subject to restriction by HUSH-MORC2, while the vast majority are kept silent by DNA methylation. We therefore considered that genome demethylation should attract the complex to activated L1 sequences. Indeed, upon loss of DNMT1 we could measure accumulation of MORC2 over activated young L1 integrants, and simultaneous CRISPRi-based depletion of DNMT1 and MORC2 led to massive accumulation of transcripts from these elements. Notably, via H3K4me3 CUT&RUN analysis we were able to show that polymorphic L1 alleles are also subject to the same restriction mechanisms. We, demonstrate that DNA methylation controls the sensitivity of the youngest L1 integrants to HUSH-MORC2 restriction and how these pathways cooperate to repress L1s.

In addition to L1 regulation, we identified the same mechanistic hierarchy between HUSH-MORC2 and DNA methylation in the transcriptional regulation of pericentromeric alpha-satellite-like repeats (ALRs). ALRs are hypomethylated, transcribed, and targeted by HUSH-MORC2 at steady-state in hNPCs. Why other repeat transcripts (e.g., SVA retrotransposons and certain LTR elements, activated in cells lacking DNMT1) remain ignored by HUSH-MORC2 requires further investigation but may be related to the CG-rich sequence composition of those elements. Details of ALR transcript structures and promoters are largely uncharacterized, but two features of ALRs: an A-rich sequence and long transcriptional units, now seem to be characteristic of HUSH-sensitive elements[30]. While functional analyses of ALRs are sparse, transcription of these arrays is a conserved property that is thought to be important to centromere function[52,53]. An ever-improving range of long-read sequencing tools promises more functional studies of ALR and other tandem repeat transcription in health and disease, and HUSH will be a key player to consider here. In this regard it is interesting that despite HUSH being vertebrate-specific, it shares structural homology with the RITS complex[21] that drives centromeric heterochromatin formation in yeast. MORC proteins are also found in basal eukaryotes and prokaryotes[54], suggesting that HUSH-like proteins with functions in silencing of repeats might form an ancient family with a conserved function in genome maintenance.

Regardless of its evolutionary origins, our observations have implications for the role of HUSH-MORC2 in hypomethylated contexts throughout human development. Predicted loss of function alleles in all members of the complex are strongly selected against according to aggregated analysis of more than 120,000 human exome sequences[55]. It was recently hypothesized that HUSH-MORC2 might provide a compensatory mechanism during the wave of DNA demethylation that occurs after fertilization[56]. In mouse embryonic stem cells (mESCs),

DNA methylation is variable depending on culture conditions and – unlike human ESCs[57] – can be removed via triple knockout of *Dnmt1*, *Dnmt3a*, and *Dnmt3b*[58]. Loss of HUSH factors caused cell cycle arrest in mESCs with low or absent DNA methylation, which was accompanied by the activation of murine L1 families[32,33]. Thus, catastrophic repeat transcription (L1s or ALRs, or other repeats in the mouse genome) and accompanying genome instability is a plausible explanation for the early developmental arrest phenotype in mice lacking TASOR[59] or MORC2[60] homologs. Whether (re-)establishment of DNA methylation at L1s during development depends on HUSH, or another H3K9me3-associated complex such as the TRIM28 system, is not clear. Nonetheless, while the repetitive mouse genome (and L1 promoter structure) is divergent from that of humans and the consequences of DNA demethylation in mouse and human ESCs are different[57], our study provides direct evidence that HUSH-MORC2 protects the human genome from repeat activation in globally-hypomethylated cells or at specific hypomethylated loci. This protective mechanism in healthy tissue may be usefully subverted in cancer: indeed, in acute myeloid leukemia and solid tumor models – both characterized by aberrant DNA methylation – depletion of MPP8 or HUSH effector SETDB1 caused activation of tumor-suppressive transposons[61,62]. Complexes between MPP8, DNMT3a and histone methyltransferases including SETDB1, have been linked to silencing of tumor suppressor E-cadherin, suggesting a dual role in H3K9me3 and DNA methylation establishment in certain contexts[63,64]. The therapeutic opportunity of HUSH-MORC2 inhibition in many cancers remains to be tested.

Finally, our observations of HUSH-MORC2 function in neural progenitors, complemented by epigenomic datasets of fetal and adult forebrain heterochromatin, have direct relevance in human neurobiology. Together with a recent study[29], we identified the HUSH-MORC2-H3K9me3 axis as a transcriptional regulator of clustered protocadherin genes. Stochastic promoter activation drives combinatorial *PCDH* expression in the nervous system, where each neuron displays a unique combination of PCDH adhesion molecules. How this beautiful genetic system is controlled by epigenetic mechanisms has been a fundamental question but is thought to involve a combination of DNA methylation and 3D structures influenced by SETDB1-dependent H3K9me3-heterochromatin[65,66]. We found that HUSH-MORC2 operates alongside DNA methylation at this locus. This mechanism may have clinical relevance: MORC2 ATPase domain mutations, which cause pronounced changes to the protein's biochemical properties and HUSH activity at transgenes[50], cause neurodevelopmental disorders[67,68]. Since PCDHs drive the formation and complexity of neuronal networks and polymorphisms at the locus are themselves associated with a range of neurodevelopmental symptoms[15], (mis-)regulation of these genes by MORC2 mutants is an attractive explanation for the etiology of MORC2-disorders, though this remains to be tested. HUSH-MORC2 restriction of demethylated L1s is also likely relevant in the human brain. We recently reported widespread, cell-type-specific transcription of evolutionarily-young L1s in the developing and adult human cortex[8], which may correlate with DNA methylation changes that occur in post-mitotic neurons[69,70]. How HUSH (and H3K9me3-heterochromatin more broadly) operates in regulation of L1s and other repeats in the brain, for example during ageing, is an exciting open question.

## Methods

### Ethics statement
The research performed on human tissue material was performed according to the highest ethical standards and in agreement with Swedish legislation. All experiments were approved by the Lund/Malmö Ethical Committee. Written consent was obtained and the participants were informed about the purpose of the research. No financial compensation was provided for tissue donors. Human fetal forebrain tissue was obtained from material available following

elective termination of pregnancy at the University Hospital in Malmö, Sweden, according to guidelines approved by the Lund/Malmö Ethical Committee under the national ethical permit (reference number: Dnr 6.1.8-2887/2017). Human post-mortem brain tissue was obtained from the University Hospital in Lund, Sweden, according to guidelines approved by the Lund/Malmö Ethical Committee, under the national ethical permit (reference number: Dnr 2019-06582, 2020-02-12). The data were anonymised and the authors did not have access to identifiable information. Tissue data were processed on a sensitive offline computer cluster with special requirements for system access.

## Antibodies

Rabbit anti-MORC2 (A300-149A, Bethyl; 1:1000 dilution), rabbit anti-TASOR (Atlas HPA006735; 1:500), mouse anti-L1-ORF1p (Millipore MABC1152, 1:1000) and HRP-conjugated anti-β-actin (Sigma A3854, 1:50,000) were used for Western blots. Rabbit anti-H3K4me3 (Active Motif 39159), rabbit anti-H3K9me3 (abcam 8898), rabbit anti-MORC2 (as above), and goat anti-rabbit IgG (abcam ab97047) were used for CUT&RUN assays (1:50-1:100 dilution). Rabbit anti-Nestin (Millipore AB5922, 1:100), goat anti-SOX2 (R&D Systems AF2018, 1:100), mouse anti-beta-III-tubulin (Biolegend 801202, 1:500), chicken anti-GFP (abcam 13970, 1:100), mouse anti-5mC (Active Motif 39649, 1:250), Alexa647 anti-rabbit (Jackson Immuno Research 711-605-152, 1:500), Alexa647 anti-mouse (Jackson Immuno Research 115-605-003, 1:500), donkey anti-goat Cy3 (Jackson ImmunoResearch 705-165-003, 1:200) and Alexa488 anti-chicken (Jackson Immuno Research, 703-546-155, 1:500) were used for immunocytochemistry.

## Cell culture

The (male) embryo-derived human neural progenitor cell line Sai2, derived from a small number of cells dissected from fetal brain tissue at Carnegie stage 16, and which has the characteristics of neuroepithelial-like stem cells[38], was cultured according to standard protocol[71]. Briefly, the cells were grown in DMEM/F12 (Thermo Fisher Scientific) supplemented with glutamine (Sigma), Penicillin/Streptomycin (1x, Gibco), N2 (1x, Thermo Fisher Scientific), B27 (0.05x, Thermo Fisher Scientific), EGF and FGF2 (both 10ng/ml, Thermo Fisher Scientific). Cells were cultured in flasks or multi-well plates (Nunc) precoated with poly L-ornithine (15μg/ml, Sigma) and laminin (2μg/ml, Sigma), fed daily with growth factors and passaged every 2-3 days using TrypLE Express enzyme (GIBCO) to detach cells and trypsin inhibitor (GIBCO) to quench the enzyme. 10 μM Rock inhibitor (Miltenyi) was added to the media following cell thawing, lentiviral transduction, and FACS. Neural differentiation followed a detailed protocol for iPSC-derived hNPCs[72]. Cultures were tested routinely for Mycoplasma infection using Eurofins Mycoplasma-check and returned a negative test.

## Lentiviral vectors

Lentiviral vectors were produced according to a standard protocol[73]. Briefly, low-passage 293T cells were grown to a confluency of ~80% and co-transfected with three packaging plasmids (pMDL, psRev, and pMD2G) plus the relevant transfer plasmid using PEI. 48 h after transfection the virus-containing supernatant was collected, filtered, and centrifuged at 25,000 x g for 1.5 h at 4 °C. The pellets were gently resuspended in PBS, aliquoted and stored at −80 °C until use. Titers were around 10^9 TU/mL, determined by qPCR.

## CRISPR approaches

To inhibit the transcription of *MORC2*, *TASOR,* and *DNMT1* individually, we followed a CRISPR interference approach described elsewhere[74]. Single gRNA sequences (listed in Table 1) were designed to bind just downstream of the relevant TSS and are listed below. Guides were inserted into a dCas9-KRAB-T2A-GFP lentiviral backbone containing both the gRNA under the U6 promoter and dCas9-KRAB-T2A-GFP

### Table 1 | Guide RNA sequences used in this study

| | 20-bp targeting sequence |
|---|---|
| LacZ control | TGCGAATACGCCCACGCGAT |
| TASOR g2 | CAAATGCGCTGCCCGGTCCT |
| MORC2 g2 | GCTTCCAAGGACCGGATCGA |
| DNMT1 g1 | TGCTGAAGCCTCCGAGATGC |
| DNMT1 g3 | TCGTCGGGCAGCGAGATGGC |

under the Ubiquitin C promoter (pLV hU6-sgRNA hUbC-dCas9-KRAB-T2a-GFP, a gift from Charles Gersbach, Addgene plasmid #71237) using annealed oligos and the BsmBI cloning site. A control lentiviral vector expressing a gRNA sequence absent from the human genome (LacZ) was used as a control in all experiments. hNPCs were transduced with an MOI of 2.5–5 and GFP-positive cells FACS-isolated (FACSAria, BD sciences) on day 10 at 10 °C. For CUT&RUN experiments, we avoided cell sorting so long as FACS analysis of the cell population indicated >90% GFP positives. For transcriptome analysis, sorted cells were pelleted at 400x*g* for 7 min, snap frozen on dry ice and stored at −80 °C for RNA isolation. Gating strategies are shown in Supplementary Fig. 10.

For simultaneous depletion of DNMT1 and MORC2 we transduced with the same dCas9-KRAB-T2A-GFP vector expressing the most efficient MORC2 gRNA, together with the pLV.U6BsmBI.EFS-NS.H2b-mCh lentiviral backbone expressing the most efficient DNMT1 gRNA and an mCherry marker. Double positive cells were FACS sorted and pellets stored at −80 °C until RNA extraction.

For DNMT1 knockout experiments, LV.gRNA.CAS9-GFP vectors were used to target the catalytic domain of DNMT1 as described elsewhere[37,57]. Lentivirus was produced as above and used to transduce hNPCs at an MOI of 10–15. The same non-targeting control (LacZ) gRNA was used for the DNMT1 KO experiment, but in the corresponding Cas9 backbone. As with CRISPRi experiments, GFP+ cells were isolated by FACS 10 days post-transduction.

## Immunocytochemistry

Cells were gently rinsed in PBS three times and fixed in 4% formaldehyde for 15 min at room temperature. For blocking, cells were incubated for 1h with 5% Normal Donkey Serum (NDS) in TKBPS (KBPS with 0.25% Triton X-100) and then incubated (overnight, 4 °C) with the primary antibody (see above) or TKPBS + 5% NDS for negative controls. The following day cells were washed twice in TKPS then once in TKBPS/NDS, then incubated (2 h, room temperature) with the relevant secondary antibody (see above) and 5 min with DAPI (1:1000, Sigma D817) as a nuclear counterstain. For 5mC staining cells were pre-treated with 0.9% Triton in PBS (15 min) followed by 2 N HCl (15 min) then 10mM Tris pH 8 (10 min) prior to incubation with the primary antibody. Following washes in KPBS, cells were imaged using a fluorescence microscope (Leica) or an Operetta CLS High Content Analysis imager (PerkinElmer).

## RNA sequencing

Total RNA was isolated using the RNeasy Mini Kit (QIAGEN) with on-column DNAse treatment. Isolated RNA was used for RT-qPCR (see below) and/or RNA sequencing. Libraries for RNA sequencing were generated using Illumina TruSeq Stranded mRNA library prep kit (poly-A selection), optimized for long fragments by reducing fragmentation time, and sequenced on a NovaSeq6000 (paired end, 2x150 bp).

**Quantification of genes.** STAR RNA-seq aligner was employed to map the paired end reads to the human genome (GRCh38), using the annotations from gencode v38. Two parameters were modified from the default in STAR v2.6.0[75], namely increasing the number of allowed multimappers for each read (--outFilterMultimapNmax 100) and

increasing the number of loci anchors (--winAnchorMultimapNmax 200). A sorted BAM was specified as the output which was further indexed using SAMtools v1.9[76]. Finally, bamCoverage v2.4.3[77] was used to create normalized tracks with a scaling factor of reads per kilobase million (RPKM). Reads from the BAM files were then quantified over the annotations from gencode v38 using subread's featureCounts v1.6.3[78], specifying a parameter to force the strandedness of each feature to be taken into consideration (-s2).

**Quantification of transposable elements.** STAR RNA-seq v2.6.0 aligner was employed to map the paired end reads to the human genome (GRCh38), using the annotations from gencode v38. To obtain uniquely mapped reads, two of the default parameters were modified, where we allow each read to map to a single locus (--outFilterMultimapNmax 1) and allowing mismatch ratio of 0.03 (--outFilterMismatchNoverLmax 0.03). The resulting BAMs were indexed using SAMtools v1.9 and then used to create coverage tracks using bamCoverage v2.4.3 where an RPKM scaling factor was implemented. Finally, the uniquely mapped reads were quantified using featureCounts v1.6.3, specifying a parameter to force the strandedness of each feature to be taken into consideration (-s2). The multimapped reads were obtained identically to the BAMs in the gene quantification process and further quantified using TEtranscripts v2.0.3[42]. To maintain consistency in our analyses, both the uniquely and multimapped reads were quantified over a curated GTF file provided by the creators of TEtranscripts[42].

**Differential expression analyses.** Differential expression analyses for all bulk RNAseq data were performed with DESeq2 v1.38.3[79] using the read count matrix from featureCounts as the input. Standard DESeq2 parameters were implemented where the counts were normalized by median of ratios. Visualization of the mean plots was achieved by obtaining the mean of normalized counts per condition, adding a pseudo count of 0.5 and log2 transforming the values.

## CUT&RUN

Three CUT&RUN protocols were used depending on the sample and target. A 'standard' protocol was used for profiling of histone modifications (H3K4me3, H3K9me3) in control or CRISPR-modified hNPC lines. A 'nuclear' protocol was used for profiling fetal and adult tissue samples. A 'crosslinked' protocol was used to profile MORC2 chromatin binding.

**Standard CUT&RUN.** We followed the protocol described elsewhere[80]. Briefly, 250-500k cells were washed twice (20 mM HEPES pH 7.5, 150 mM NaCl, 0.5 mM spermidine, 1x Roche cOmplete protease inhibitors) and attached to 10 μL ConA-coated magnetic beads (Bangs Laboratories) that had been pre-activated in binding buffer (20 mM HEPES pH 7.9, 10 mM KCl, 1 mM CaCl₂, 1 mM MnCl₂). Bead-bound cells were resuspended in 50 mL buffer (20 mM HEPES pH 7.5, 0.15 M NaCl, 0.5 mM Spermidine, 1x Roche complete protease inhibitors, 0.05% w/v digitonin, 2 mM EDTA) containing primary antibody (see above) and incubated at 4 °C overnight with gentle rotation. Beads were washed thoroughly with digitonin buffer (20 mM HEPES pH 7.5, 150 mM NaCl, 0.5 mM Spermidine, 1x Roche cOmplete protease inhibitors, 0.05% digitonin). After the final wash, pA-MNase (a generous gift from Steve Henikoff) was added in digitonin buffer and incubated with the cells at 4 °C for 1 h. Bead-bound cells were washed twice, resuspended in 100 μL digitonin buffer, and chilled to 0–2 °C. Genome cleavage was stimulated by addition of 2 mM CaCl2 at 0 °C for 30 min. The reaction was quenched by addition of 100 mL 2x stop buffer (0.35 M NaCl, 20 mM EDTA, 4mM EGTA, 0.05% digitonin, 50 ng/mL glycogen, 50 ng/mL RNase A, 10 fg/mL yeast spike-in DNA (a generous gift from Steve Henikoff)) and vortexing. After 10 min incubation at 37 °C to release genomic fragments, cells and beads were pelleted by centrifugation (16,000 × g, 5 min, 4 °C) and fragments from the supernatant were purified by PCR clean-up spin column (Macherey-Nagel).

**Crosslinked CUT&RUN.** The procedure was the same as above except for the following changes: 900K-1M cells were used and crosslinked for 1 min using 1% formaldehyde (16% solution, Thermo Scientific) diluted in media. Cells were washed three times (20 mM HEPES pH 7.5, 150 mM NaCl, 0.5 mM spermidine, 1x Roche complete protease inhibitors, 1% Triton X-100, 0.05% SDS) with a slightly increased centrifugation speed to avoid pellet loss. After attachment to ConA beads, the mixture was resuspended in buffer (20 mM HEPES pH 7.5, 150 mM NaCl, 0.5 mM spermidine, 1x Roche complete protease inhibitors, 1% Triton X-100, 0.05% SDS, 0.05% digitonin, 2 mM EDTA) containing primary antibody. Following 4 °C overnight incubation with primary antibodies, beads were washed thoroughly with digitonin buffer (20 mM HEPES pH 7.5, 150 mM NaCl, 0.5 mM Spermidine, 1x Roche complete protease inhibitors, 1% Triton X-100, 0.05% SDS, 0.05% digitonin). After incubation at 37 °C to release genomic fragments, crosslinking was reversed by adding 0.09% SDS and 0.22 mg/ml proteinase K (Thermo Scientific) and incubation overnight at 55 °C. The following day, DNA was purified by spin column as above.

**Nuclear CUT&RUN.** Isolation of nuclei from embryonic and adult human brain tissue was performed according to an established protocol[81]. In brief, tissue was dissociated in ice-cold lysis buffer (0.32 M sucrose, 5 mM CaCl₂, 3 mM MgOAc₂, 0.1 mM Na₂EDTA, 10 mM Tris-HCl, pH 8.0, 1 mM DTT) using a 1-mL Dounce homogenizer (Wheaton). The homogenate was carefully layered on a sucrose cushion (1.8 M sucrose, 3 mM MgOAc₂, 10 mM Tris-HCl, pH 8.0, and 1 mM DTT) before centrifugation (30,000 × g, 2 h 15 min). Pelleted nuclei were softened for 10 min in 100 mL of nuclear storage buffer (15% sucrose, 10 mM Tris-HCl, pH 7.2, 70 mM KCl, and 2 mM MgCl₂) then resuspended in 300 mL of dilution buffer (10 mM Tris-HCl, pH 7.2, 70 mM KCl, and 2 mM MgCl₂) and run through a cell strainer (70 mm). Cells were run through the FACS (FACS Aria, BD Biosciences) at 4 °C at a low flow rate using a 100 mm nozzle (reanalysis showed >99% purity). For isolation of NeuN+ nuclei (adult post-mortem samples only), nuclei were incubated with AlexaFluor 488 anti-NeuN (abcam 190195, 1:500) for 30 min on ice. 300k AlexaFluor488-positive nuclei were sorted per tissue piece. The sorted nuclei were pelleted at 1300 × g for 15 min and resuspended in 1 mL of ice-cold nuclear wash buffer (20 mM HEPES, 150 mM NaCl, 0.5 mM spermidine, 1x cOmplete protease inhibitors, 0.1% BSA) and 10 μL per antibody treatment of ConA-coated magnetic beads (Epicypher) added with gentle vortexing. All buffers contained 0.1% BSA and tips were pre-treated with 0.1% BSA.

**CUT&RUN analysis.** Illumina sequencing libraries were prepared using the Hyperprep kit (KAPA) with unique dual indexed adapters (KAPA), pooled and sequenced on a Nextseq500 instrument (Illumina). Paired-end reads (2x75) were aligned to the human genome (hg38) using bowtie2 (−local −very-sensitive-local −no-mixed −no-discordant −phred33 -I 10 -X 700) and then converted converted to bam files using samtools[76,82]. To extract the uniquely aligned reads, BAM files were filtered setting a threshold for mapping quality of 10 (MAPQ10). Coverage bigwig tracks were created using bamCoverage (deepTools)[77] and normalized with a rounds per kilobase million (RPKM) scaling factor. The coverage tracks displayed in all figures were visualized by the genome browser IGV. Peaks were called using HOMER findPeaks v4.10[83], searching for peaks with variable lengths using the histone command (-style histone). In order to capture both baseline and knockdown specific peaks, we used the command for both conditions (control and CRISPRi) and filtered out all non-canonical chromosomes. H3K9me3 peaks were also filtered according to length, where all peaks shorter than 1 kb were excluded from the analysis. Finally, to make a comprehensive peak list, we concatenated

**Table 2 | RT-qPCR primer sequences used in this study**

|  | Forward (5'-3') | Reverse (5'-3') |
|---|---|---|
| MORC2 | ACATGAAGACGCAGGAAGAG | ACTTCCAAGGGCAATTTCTT |
| DNMT1 | GATCGAGACCACGGTTCCTC | CGGCCTCGTCATAACTCTCC |
| TASOR | TGAAGACATTGCAGGTTTCATTC | CATCCAGGCTATCAACACCAG |
| GAPDH | TTGAGGTCAATGAAGGGGTC | GAAGGTGAAGGTCGGAGTCA |
| HPRT1 | ACCCTTTCCAAATCCTCAGC | GTTATGGCGACCCGCAG |
| ACTB | CCTTGCACATGCCGGAG | GCACAGAGCCTCGCCTT |

and merged the baseline and knockdown specific peaks. Tag directories were created with HOMER's makeTagDirectory v4.10 using the standard parameters. Finally, we obtained the differential enrichment over the peaks discovered by SEACR using HOMER's getDifferentialPeaks setting a fold change threshold of 3 (-F 3) and keeping the default settings for the rest of the parameters. Heatmap matrices were computed using DeepTools' computeMatrix v2.5.4 and later visualized with plotHeatmap 2.5.4 from the same deepTools package[77].

## RT-qPCR
cDNA was generated by reverse transcription of 500ng RNA with random hexamer primers and Superscript III (Invitrogen) and analyzed by RT-qPCR with SYBR Green I master (Roche) on a LightCycler 480 instrument (Roche). Data are represented with the ΔΔCt method normalized to housekeeping genes *ACTB* and *HPRT*. Primer sequences are listed in Table 2.

## Western blot
Cells were lysed in RIPA buffer (Sigma-Aldrich) containing complete protease inhibitor cocktail on ice for 30 min and then pelleted (20 min, 17,000 x *g*, 4 °C). Supernatants were collected and mixed with Novex LDS 4x loading dye containing reducing agent (Thermo) boiled at 95 °C for 5 min before being separated on a 4–12% Tris-glycine SDS-PAGE gel (200 V, 45 min). Proteins were transferred from the gel to a PVDF membrane using the Transblot-Turbo Transfer system (BioRad). The membrane was then washed twice (15 min) in Tris-buffered saline with 0.1% Tween (TBST) and blocked for 1 h in TBST with 5% skimmed milk (MTBST) before incubation (overnight, 4 °C) with the primary antibody diluted in MTBST. The following day the membrane was washed twice in TBST (15 min) and incubated (1 h, room temperature) with HRP-conjugated anti-mouse or anti-rabbit secondary antibody (Santa Cruz Biotechnology, 1:5,000) diluted in MTBST. After washing the membrane twice in TBST and once in TBS, protein was detected by chemiluminescence using ECL Select reagents (Cytiva) as per manufacturer's instructions and imaged on a Chemi-Doc system (BioRad). The membrane was stripped using the Restore PLUS Wester Blot Stripping Buffer (Thermo) as per instructions, re-blocked for 1 h in MTBST after which the procedure for the β-actin staining was performed as above.

## Oxford Nanopore whole-genome DNA sequencing
High molecular weight DNA was extracted from frozen pellets (1 million cells) using the Nanobind HMW DNA Extraction kit (PacBio) following the manufacturer's instructions. The final product was eluted in 100 µL. DNA concentration and quality were measured using Nanodrop and Qubit from the top, middle, and bottom of each tube, and by agarose gel electrophoresis. Further shearing was done with a Megaruptor (Diagenode) at speed 31. Whole-genome sequencing was done using the SQK-LSK109 Ligation Sequencing kit (Oxford Nanopore Technologies) and FLO-PRO002 PromethION Flow Cell R9 Version on a PromethION (Oxford Nanopore Technologies) at SciLifeLab National Genomics Infrastructure in Uppsala. Basecaller version ont-guppy-for-promethion 6.2.11 was used for basecalling. Reads passing QC had an average length N50 of 13.99 kb, and were aligned to the human genome (hg38) using minimap2 (-a -x map-ont)[84] giving an approximate 40x genome coverage for each sample (10.9M reads for control hNPCs, 8.5M reads for MORC2-CRISPRi hNPCs). The program Transposons from Long DNA Reads (TLDR) was used to call polymorphic insertions[39]. 5mC base modifications were detected using the nanopolish tool[85] The program methylartist was used to create methylation databases (methylartist db-nanopolish -m nanopolish.tsv.gz -d nanopolish.db) and for all subsequent visualizations[86].

## Oxford Nanopore cDNA sequencing
1.5 µg total RNA was isolated as above for each of the four samples tested (Control, MORC2-CRISPRi, TASOR-CRISPRi and DNMT1-KO hNPCs, 10 days post-transduction in each case). Ribosomal RNA was depleted using the RiboMinus Eukaryote System v2 (ThermoFisher). 5 ng of the resulting RNA was used to make barcoded cDNA libraries (2 barcodes per sample) using the cDNA-PCR kit SQK-PCB-111-24 (Oxford Nanopore Technologies), which incorporates a strand-switching step to enrich for full-length transcripts. Fifteen PCR cycles were used with 7 min extension time. Inspection of the final cDNA product by capillary electrophoresis confirmed a peak at around 2-2.5 kb, approximately the size of an average human mRNA. The libraries were sequenced with a FLO-PRO002 PromethION Flow Cell R9 Version on a PromethION (Oxford Nanopore Technologies) at SciLifeLab National Genomics Infrastructure in Uppsala. Base calls were done using Guppy 6.1.5 and reads mapped using minimap2 (-a -x splice) to the hg38 genome or to the L1HS consensus sequence. More than 90% of reads mapped in all cases. The number of mapped reads per sample were: 16.7M (Control); 20.6M (MORC2-CRISPRi); 16.6M (TASOR-CRISPRi) and 17.6M (DNMT1-KO). Alignments were visualized using IGV using a mapQ threshold of 10.

## Statistics and reproducibility
One sample of MORC2 CRISPRi hNPCs was excluded from transcriptome analysis due to poor RNA quality. CUT&RUN and Western blot experiments were performed at least twice with similar results. Transcriptomes were obtained with at least three independent samples per condition. Transcriptome data for TASOR (*n* = 4) and MORC2 (*n* = 3) CRISPRi experiments were pooled as the treatment group to assess changes over common targets, and to control for potential off-target effects. Nanopore genome sequencing experiments were done once per sample. Polymorphic L1 elements were called using whole genome sequencing data from two samples – control and MORC2 CRISPRi hNPCs – in the same genetic background. No statistical methods were used to pre-determine the sample size, experiments were not randomized, and the investigators were not blinded to allocation during experiments and outcome assessment.

## Reporting summary
Further information on research design is available in the Nature Portfolio Reporting Summary linked to this article.

## Data availability
There are no restrictions on data availability. The RNA and DNA sequencing data presented in this study have been deposited at GEOs: GSE242143 bulk RNA-seq, CUT&RUN and ONT whole genome sequencing of hNPC CRISPRi and control cells; GSE224747 3' single nuclei RNAseq, CUT&RUN and bulk RNAseq of fetal tissue samples; GSE209552 3' single nuclei RNAseq of adult post-mortem samples; GSE211871 Adult post-mortem NeuN+ CUT&RUN sequencing. Source data are provided with this paper.

## Code availability
Code has been deposited on GitHub (https://github.com/NinoPandiloski/DNA-methylation-governs-the-sensitivity-of-repeats-to-restriction-by-the-HUSH-MORC2-corepressor).

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

## Acknowledgements

We thank M. Persson-Vejgården, A. Hammarberg, Sol da Rocha Baez, and E. Monni for their technical assistance, S. Henikoff for the gift of the pA-MNase used in CUT&RUN experiments and A. Falk for providing hNPC lines. We thank D. O'Carroll, D. Prigozhin, G. Faulkner,

and Y. Modis for comments on the manuscript. We acknowledge the Cell and Gene Therapy Core at Lund University for assistance with CRISPRi guide design and the support of the National Genomics Infrastructure (NGI)/Uppsala Genome Center and UPPMAX for providing assistance in massive parallel sequencing and computational infrastructure. Work performed at NGI/Uppsala Genome Center has been funded by RFI / VR and Science for Life Laboratory, Sweden. The authors would like to acknowledge Clinical Genomics Lund and the Center for Translational Genomics (CTG) at Lund University for also providing expertize and service with sequencing and analysis. This work was supported by the Swedish Government Initiative for Strategic Research Areas (MultiPark & StemTherapy) and grants from the Swedish Research Council (VR, 2018-02694 to J.J., 2020-01660 to Z.K. and 2021-03494 to C.H.D.), the Swedish Brain Foundation (Hjärnfonden, FO2019-0098 to J.J., FO2022-0079 to Z.K and FO2023-0229 to C.H.D.) and the Swedish Society for Medical Research (SSMF, S19-0100 to C.H.D.). This project has been made possible in part by grant 2023-331773 from the Chan Zuckerberg Initiative DAF, an advised fund of the Silicon Valley Community Foundation. We also acknowledge the Crafoord Foundation, Hedlunds Foundation, and Jeanssons Foundation (project grants to C.H.D.).

## Author contributions

N.P., J.J., and C.H.D. conceived and designed the study. V.H., O.E.K., S.K., F.D., G.C., J.M., D.A., J.G.J., and C.H.D. performed experimental work. N.P., S.K., P.G., and R.G. performed bioinformatic analyses. M.E.J., A.A., E.E., and Z.K. contributed essential materials or expertize. C.H.D. wrote the manuscript with assistance from N.P. and V.H., who prepared the figures. J.J. and C.H.D. supervised the study and acquired funding. All authors reviewed the final version.

## Funding

## Competing interests

The authors declare no competing interests.
