## [Peer Review File · Nature Communications]

DNA methylation governs the sensitivity of repeats to restriction by the HUSH-MORC2 corepressorREVIEWER COMMENTS

Reviewer #1 (Remarks to the Author):

This manuscript explores the interplay between DNA methylation and the HUSH complex, and its effector protein MORC2, in mediating retrotransposon repression in a hNPC somatic cell model. The topic is of very high interest. I congratulate the authors on the quality of the CRISPRi experiments and steps taken to measure retrotransposon expression and repression robustly with RNA-seq and CUT&RUN. These are challenging experiments and for the most part they are executed well here. However, I found the manuscript as it stands confusing in parts. I think it would benefit from being more organised conceptually and more considered in its comparisons to the existing HUSH literature, and I feel limited additional experimental data are required to confirm one of the main conclusions.

Comments:

1) If I was to summarise the main point (please correct me if I'm wrong) it is that HUSH does not seem to be doing much (Fig. 2) to target L1 in the hNPCs, unless DNMT1 is depleted, which triggers minor L1 transcription leading to HUSH recruitment (dovetailing with the Paul Lehner lab model from reference 30). HUSH depletion in this environment of minor L1 expression then leads to major L1 expression. In that case, how relevant are the findings from Fig. 2? They show MORC2 was at some point recruited to older, potentially 5' truncated L1s lying in sense to gene introns. That seems at odds with the reference 30 model of recruitment to intronless, independently transcribed L1s, yes? I don't think it is sufficient to explain away this difference based on the methylation landscape of the cells involved either. The hNPCs here have a mean L1 methylation of ~80% and HEK293T and HeLa cells (used in ref 30) have a mean L1 methylation of >75% (ref. 49 Fig. S2). Please consider this reasoning and clarify the significance of Fig. 2 and its relationship to the current model of HUSH silencing.

2) I appreciate the authors cite reference 59 in the Discussion but I feel this is inadequate and it should be covered in the Introduction. This prior work from Helen Rowe's lab showed that the impact of HUSH depletion on retrotransposon repression was methylation dependent (mESCs tested in 2i vs serum). This affirms one of the main conclusions of the present work.

3) lines 81 and 389 - cancer cell lines may have aberrant methylomes but, to be fair to those models, the hNPCs used here appear very clonal in their methylomes too (based on the plots in Fig. 3G). Their relevant levels of genome-wide methylation arguably resemble those of the workhorse cancer cell lines used to study HUSH thus far. Can the authors point to methylomes from fetal brain tissue to show the hNPC cell line faithfully recapitulates both the level and heterogeneity of methylation observed throughout the genome? If not, it may not be the case that the hNPC methylome explains differences to HUSH data obtained from cancer cell lines.

4) The title for Fig. 1 is at odds with what is presented, there is no data for retrotransposons in the figure that I can tell.

5) L1Hs methylation is high genome-wide in the hNPCs, but do the authors observe the "escapee" L1 loci? As observed elsewhere, these should have unusually low methylation in the hNPCs (PMID: 30692270, ref 2 and ref 38). Whether these full-length L1s have a YY1 site or not

(line 228) they can be transcribed (and retrotranspose) and therefore should attract MORC2 and be marked by H3K4me3 in the control hNPCs. Some of these escapee loci are listed in the supplemental of reference 2. The violin plots in Fig. 3B suggest there are a reasonable number of such escapee L1s in the control hNPCs (and they could be identified from the methylartist segmeth output of the ONT data processing step mentioned on line 692).

6) line 202 - the mappability of the youngest L1s (L1Hs) with RNA-seq is arguably overstated here. Other works (summarised in PMID: 32576954) have suggested transcription of these loci is very difficult to measure with short-read RNA-seq. This is one of the reasons why I commend the authors on the use of H3K4me3 in their study but also request they perform H3K4me3 CUT&RUN on MORC2/DNMT1 CRISPRi cells. These data are important to corroborate the analyses presented in Fig. 4, which is the heart of the study, and add to the profiles in Fig. 4G.

7) An observation the authors may wish to track down: both of the odd L1s in Fig. 3G are marked by ENCODE as enhancers (enhD). Perhaps their low methylation is a reflection of a regulatory role?

8) Why is the MORC2 binding so weak in the examples provided in Fig. 4G compared to Fig. 2H? If HUSH is being recruited to these sites by the active transcription of the L1 promoter presumably one would observe more MORC2? (another reason to do H3K4me3 on the MORC2/DNMT1 CRISPRi cells, to confirm these as sites).

9) line 407: this sentence could use a qualifier as at present the evidence presented does not demonstrate independence of DNA methylation and HUSH-MORC2 restriction.

10) It would be helpful to show some full-length L1Hs loci from Fig. 3E in the style of the ONT and CUT&RUN profiles from Fig. 3G. If these can indeed be mapped uniquely with RNA-seq then presumably clear examples in support of the genome-wide trends shown in Fig. 3D will be found?

11) For the ONT analysis, as shown in Fig. 3B, where were the coordinates of L1Hs copies obtained from, and did these cover only 5'UTR regions or the entire L1Hs (which tends to underestimate changes due to L1 body methylation being resistant to change).

Note: I previously commented on a preprint version of this work on social media, which I see has been acknowledged in the manuscript. I don't see this as an issue in providing a formal review but readers may wish to be aware of it.

Geoff Faulkner (University of Queensland)

Reviewer #2 (Remarks to the Author):

In this study, Pandiloski et al address a critical knowledge gap related to the interplay between the HUSH complex and DNA methylation in the regulation of genomic repeats, specifically LINE-1 retrotransposons. While it is known that HUSH binds LINE-1 transcripts and recruits SETDB1 and MORC2 to repress chromatin, the specific relationship between HUSH and DNA methylation - a central regulator of repeat transcription - has remained largely unexplored.

Previous studies into the role of the HUSH complex in silencing retroelements often used models with low levels of DNA methylation. To overcome this limitation, the researchers turned to human neural progenitor cells (hNPCs), which exhibit high levels of DNA methylation but can tolerate the depletion of DNMT1 and therefore study the role of the HUSH complex in both scenarios within the same system using CRISPRi. This approach was successfully executed in this study.

Their key conclusions are that even in the absence of MORC2 or the HUSH subunit TASOR in hNPCs, LINE-1 retrotransposons remain silenced due to robust promoter methylation. However, when genome demethylation occurs upon depletion of DNMT1, and evolutionarily young LINE-1 elements become active, MORC2 binding, and presumably HUSH component binding increases to these regions. They suggest that silencing is therefore layered and that simultaneous depletion of DNMT1 and MORC2 results in a substantial accumulation of LINE-1 transcripts (higher than in the single depletions).

This observation aligns with previous research conducted in models such as mouse embryonic stem cells and cancer cell lines, which have low levels of DNA methylation. In these models, deletion of the HUSH complex leads to increased LINE-1 expression and a type I interferon response. The researchers also extend their study to pericentromeric α -satellites and clustered protocadherin genes, repetitive elements that play important roles in chromosome structure and neurodevelopment, with the latter being previously reported during mouse brain development and in brain organoids.

This paper is important and sheds light on the critical functions of HUSH and MORC2 in the context of low DNA methylation levels in development. It contributes to the growing body of knowledge regarding the intricate mechanisms governing epigenetic control during development and in somatic cells. However, as discussed below I did have concerns with their model and would recommend publication if these can be addressed.

Major issue

Their general model is that there are two independent layers of regulation and that DNA demethylation via loss of DNMT1 sensitizes L1 elements to HUSH-MORC2 restriction (Figure 4H). However – for the reasons stated below this does not completely make sense to me.

158 Following MORC2 and TASOR depletion experiments they observed loss of H3K9me3 over hundreds of sites, confirming the HUSH pathway as functional in their hNPC model. Furthermore, cut and run experiments showed MORC2 bound to the majority of these sites. This is what would be expected for HUSH-dependent loci. However – it is then difficult to reconcile these results with the next section of their manuscript: On line 210 they don't see a major transcriptional deregulation of L1s in hNPCs lacking MORC2 or TASOR – nor an effect on K4 but they don't report the effect on K9. They find that the loss of DNMT1 does not affect H3K9me3 deposition (Supp Fig 3C), which presumably implies that there was K9 at these loci – but they haven't shown that it is MORC2/TASOR dependent (or not). This would seem to be essential as the basis of their subsequent model relies on the fact that in the next section: 'DNA methylation status controls L1 sensitivity to HUSH-MORC2 restriction' they explain that 'loss of DNA methylation activates transcription of young FL-L1s, which become targets of MORC2 (Fig.

4G,H).’ At first glance this seems to make sense – but if prior to the release of DNMT1 they were not already targets of MORC2/TASOR (which requires active transcription) – how did those loci acquire K9 - if they were not transcriptionally active i.e. it seems to become a ‘chicken and egg’ situation – that requires an explanation – nor are the two ‘independent’ as they seem to suggest?

To expand on this further:

The section:

Line 255 DNA methylation status controls L1 sensitivity to HUSH-MORC2 restriction is important but is written and presented in a confusing manner –it would benefit the reader if they were able to present a clearer flow of events

Line 265: Firstly, they don’t interpret Fig 4A – and simply say: ‘Upon depletion of DNMT1 we could measure accumulation of MORC2 over activated, young FL-L1s (Fig. 4A).’ I’m not sure how to interpret the right-hand panels of Fig 4A? and – what would happen to MORC2 in the absence of TASOR? As discussed above, It’s unclear as to how much MORC2/TASOR/K9 is already present at these L1 elements in the presence of DNMT1 (ie at steady state) – as more appears to be recruited in the absence of DNMT1? This is important as the reader needs to know whether these are loci which would (i) be normally classified as ‘HUSH-dependent loci’ – or (ii) do they ‘become HUSH dependent following the loss of DNA methylation’, as they become transcriptionally active and recruit HUSH/ MORC2 components?

One would presume it is in fact the former – as they should have been ‘HUSH-defined’ by the presence of TASOR/MORC2-dependent deposition of K9. However, if this is indeed the case, how is the K9 deposited if HUSH components are not present and they are not transcriptionally active? This all needs to be clearly explained to help understand their model. For example in Fig 4G – there is increased gene expression following depletion of DNMT1, with MORC2 recruited. Furthermore, derepression becomes most marked following the double depletion. But for these genes they don’t show the levels of K9 deposition. This is confusing as either (i) there is no K9 present in the control – in which case how could they initially define this gene as ‘HUSH-dependent’ or (ii) there is K9 deposited – which is also confusing – as how does it get deposited in the absence of transcription and in the absence of MORC2/TASOR recruitment to that locus?

In fact the title of the previous paragraph section (DNA methylation, but not HUSH-MORC2, controls L1 transcription in hNPCs) is misleading as it gives the strong impression that L1 silencing is maintained by promoter DNA-methylation and that in the absence of MORC2/TASOR, at least for the majority of L1 elements, silencing is maintained i.e. that MORC2/TASOR are irrelevant for silencing these L1 elements (yet they have K9?) (The fig legend for Fig 3 reads: ‘FL-L1s are generally silenced by DNA methylation but not HUSH-MORC2 in hNPCs’. However, in the following paragraph they go on to show that the effect is clearly much more nuanced – that in fact there is a layering effect, and that there is only maximal release of L1 element repression following the double depletion – i.e. DNMT1 and MORC2. This latter point makes sense and is logical, so it makes the first title incorrect? Both the title and the content of the previous paragraph (DNA methylation, but not HUSH-MORC2, controls L1 transcription in hNPCs) needs rephrasing – so as not to confuse the reader that L1 silencing is all about DNA promoter methylation?

They subsequently contradict their previous title by summarizing their work: 285 ‘Simultaneous loss of DNMT1 and MORC2 therefore causes a massive accumulation in the levels of young FL-L1 transcripts, suggesting independent transcriptional and co/post-transcriptional L1 control by the two epigenetic pathways.’

ALRs

They could take the opportunity to emphasise that, presumably because the ALRs are hypomethylated, the effect of MORC2/TASOR depletion is more marked here than in other regions of NPCs and in fact shows a more major effect than the DNMT1 deletion? – in fact they don’t really mention the effect of the loss of MORC2/TASOR here at all?

Their statement L325 that: DNA methylation status was unchanged over ALRs in cells depleted of MORC2 – suggesting that the pathways operate independently, as was observed at L1s is not necessarily correct. If DNA methylation machinery was recruited by HUSH components (as is certainly possible) – then the loss of MORC2 would indeed prevent silencing, but HUSH would still likely be recruited i.e. DNA methylation might still be dependent on core HUSH components – if not directly dependent on MORC2 – so the two would then be interdependent and not independent?

General impression is that the ALRs are more HUSH sensitive because they are hypomethylated – why are they hypomethylated?

Other points:

Introduction

DNA methylation was examined in the context of an SVA reporter by Robbez-Masson, L et al 2019 and its recruitment was HUSH dependent so this paper should be cited early

67 The human silencing hub (HUSH) complex²¹ has emerged as an important epigenetic
68 regulator of repeats through H3K9me3.^{7,22–24} - The correct references here should be 21 –
24.

Line 70

Recent data have shown that HUSH also participates as an adapter for cotranscriptional RNA processing complexes, leading to the destruction or termination of targeted transcripts.^{26–28} Such an RNA-dependent mechanism is reminiscent of evolutionarily ancient systems such as the yeast RNA-induced transcriptional silencing⁷³ (RITS) complex, and indeed analysis of HUSH protein architectures revealed striking similarities with RITS.²²

The combination of these two sentences doesn’t seem logical – it reads like a non-sequitur. The first relates to the link between HUSH and the destruction/termination of transcripts (NEXT complex). The second relates to the silencing mechanism of HUSH which is indeed evolutionarily related to RITS, as both are RNA-dependent silencing. The link between the two is not about how HUSH relates to the degradation machinery?

Results

Line 167 – should note reference 30

How representative is the NPC cell line – was it clonal in origin – more details would be helpful here?

Several additional elements could be included to further validate the system; it would be helpful to incorporate other markers that characterise the identity of the cell line, such as Sox1, Sox2, Musashi-1. Moreover, protein expression of the HUSH complex (by western blotting or immunofluorescence) would be a helpful addition to the 2019 proteomic data. When presenting the genome-wide DNA methylation plot in Figure 1D, it would be valuable to dissect it in some of the repetitive sequences - this could provide insights into whether these sequences are completely methylated and to study and discuss the evasion of DNA methylation in certain LINE-1 elements as reported previously in different studies. Can they address the temporal discrepancy between the snRNAseq analysis, which covers the period of 7.5-10.5 weeks post-conception, and the characterization of the hNPCs, which are at 6 weeks post-conception. The same is seen with CUT&RUN profiling of H3K4me3 and H3K9me3 in human fetal forebrain samples at 7- and 10-weeks post-conception. Discussing how these factors change during embryonic development would provide a more comprehensive understanding of the research and clarify these points.

The authors emphasise that MES cells and cancer cell lines may show incomplete or aberrant DNA methylation – are the methylation patterns of their hNPCs representative of normal somatic cells? How does 80% genome wide CpG methylation relate to other cell types?

Do the accumulated L1 transcripts have any impact on cell fitness, neural differentiation capacity, or any phenotypic outcomes? The same question applies to the dysregulation of ALRs and protocadherin genes. For instance, while the upregulation of the IFN response is demonstrated in individual CRISPRi experiments, it is not shown for the double depletion of DNMT1 and MORC2. Is it also increased in the double depletion compared to the single ones?

Fig 2 – I was a bit confused by the MORC2 locus – as Douse has previously shown that the MORC2 locus is already HUSH repressed – or at least that there is K9 deposited in a HUSH dependent manner? –

So on Line 149, it might be worth pointing out that the endogenous MORC2 locus, unlike the TASOR locus, is known to be HUSH repressed – as shown in Figure 2B where there is K9 present?

However, I was then surprised that in figure 2D they do not see an increase in MORC2 protein expression following TASOR depletion?

Fig 2E – I wasn't clear what information I was supposed to gain from this figure?

Fig 2F – the legend says: 'Volcano plot showing H3K9me3 changes genome-wide over H3K9me3 peaks in

179 MORC2 (n=2) and TASOR (n=2) CRISPRi hNPCs compared to controls (n=4) as measured by 180 CUT&RUN epigenome profiling 10 days post-transduction (significant points defined by fold-change>3 and Poisson p-value <1e-04).

It wasn't clear how this experiment was performed – have they just pooled data from MORC2 and TASOR depletions? Why aren't they showing the differential effect of a MORC2 and a TASOR

depletion – are we to assume that in the absence of MORC2 or TASOR the same genes are affected? it would be clearer to show separately and clarify that they behave similarly and how different the double depletions are.

Lines 163 – 168 The data shown in Fig 2I is a striking observation and supports the model suggested in ref 30 and should be attributed as such in the text? The authors discuss that HUSH-dependent silencing primarily targets young active L1s retrotransposons located on the same DNA strand as the host genes; however, in multiple figures such as in figure 4G the authors show antisense copies to the host gene. It would be helpful to clarify why these examples were chosen.

Line 160 they show MORC2 bound to the majority of these sites – can they be more quantitative than this.

Doesn't Fig 4D needs a control CRISPRi to see if there is any effect of TASOR/MORC2 depletion and can they show a statistical analysis of these results? Maybe their data is already analysed relative to the control – in which case this should be clearly stated.

Lines 224 – 233

Was there anything significant about the 19 FL-L1s which were upregulated with HUSH loss – do they know why these loci were aberrantly DNA-methylated?

Line 226 – The significance of this statement is somewhat unclear:

Interestingly, upon examination of the promoters of the three candidate elements in our Nanopore sequencing dataset we could detect a proportion of demethylated reads (Fig 3G and Supp. Fig. 3E), suggesting that these elements are incompletely methylated in hNPCs.

– I assume they are implying that these sites weren't methylated and therefore needed MORC2/TASOR for repression – but it would be helpful to know if this was unique to these sites – are they suggesting that all the young L1s which were not HUSH-sensitive are fully methylated?

Discussion

413 Why other repeat transcripts (e.g. SVA retrotransposons and certain LTR elements, activated in cells lacking DNMT1) remain ignored by HUSH MORC2 remains an open question.

Actually – It's not such an open question – they don't fulfill the criteria for HUSH-dependent repression

Reviewer #3 (Remarks to the Author):

In the manuscript entitled “DNA methylation governs the sensitivity of repeats to restriction by the HUSH-MORC2 corepressor”, Pandiloski et al explore the role of the HUSH complex in silencing repeat sequences (primarily L1s) in human neural progenitor cells and its interplay with DNA methylation machinery. This is relevant because, as the authors note, the majority of

studies that have explored L1 regulation by HUSH have been done in cells (K562, ESCs) where DNA methylation is strongly reduced, compared with most somatic cell populations where CpG methylation is much higher. Using an elegant CRISPRi approach against TASOR (a core HUSH component) and MORC2 (a peripheral HUSH component and ATPase) the authors show that a subset of L1 elements are targeted by HUSH to establish H3K9me3. These elements share some unique features, they are enriched in genes, enriched in the sense orientation, and display low levels of transcription and hypomethylation. L1 elements are not broadly activated however upon MORC2 or TASOR loss, unlike upon Dnmt1 loss, where massive L1 activation and promoter H3K4me3 appearance are observed. Interestingly, upon Dnmt1 loss, MORC2 is more strongly recruited to young L1s, and this recruitment is important for reducing the accumulation of L1 transcripts, as there is a synergistic effect on L1 levels upon Dnmt1 and TASOR/MORC2 loss. Finally, the authors observe that pericentromeric ALR repeats, and members of the PCDH gene family, are regulated by HUSH, and like L1s share some similar features, namely they are somewhat hypomethylated relative to most of the genome, and are regulated by a combination of DNA methylation and HUSH.

The manuscript is generally well written and likely to be of broad interest across fields of evolution, neurobiology, genomics, and chromatin biology. The authors have developed a useful model to query the interplay between global DNA methylation and sensitivity to HUSH-MORC2 repression and the presented data adds to the mechanistic understanding of HUSH complex mediated regulatory control of endogenous repeats. This study adds to the growing body of evidence that the HUSH complex functions to dampen the expression of expressed repeat sequences and clarifies the relationship between DNA methylation and HUSH mediated control of L1 elements.

Some comments for the authors consideration:

Figure 2F: It is unclear what the dots represent, are these peak regions? Individual repeats? Small windows of the genome? I get that they represent areas where H3K9me3 is either changed or unchanged, but it is unclear how they were defined. Furthermore, it is not immediately clear how the data shown in Figure 2F was generated as the description of “pooling replicates” in the results (line 157-158) is a bit vague. A brief explanation in the results would help the reader follow the criteria used to identify the 532 methylated regions identified as lost in CRISPRi. You could consider adding a panel to Figure 2 or supplemental – for example is the fold change correlated between the two KD datasets for the 532 methylated regions identified as lost in both?

Figure 2G: As LINE elements are more abundant than LTRs in the genome, it may be most appropriate to provide information on relative enrichment rather than absolute numbers.

Figure 2I: Is expression of the genes flanking the 181 HUSH-targets altered? (is this the 139 HUSH-bound genes shown in Supplemental 3A? if so why 181 vs 139?)

Figure 4D-F: My assumption is that the L1s affected by DNMT1/MORC2 loss do not show the same bias as those shown in figure 2. Can the authors make a figure that shows that upon DNA methylation, MORC2 affects L1s independent of their position/orientation within genes?

Figure 5: I suggest providing methylation data for the protocadherin locus as in Figure 3

(presuming this data is readily available in the existing nanopore dataset).

Figure 5F. In this zoomed out view, it appears MORC2 may have a few binding sites within and near the PCDHgamma cluster. Can the authors show zoomed up views at these peaks? Are these peaks overlapping L1s? The authors mention that MORC2's strongest peak is at the TAF7 gene but this is not labeled. Do they authors have a model for how MORC2 is recruited here (beyond that it is a single exon gene)? Is this a pseudogene or an otherwise unique transcript? Is it expressed in NPCs or regulated by DNA methylation?

Manuscript: "DNA methylation governs the sensitivity of repeats to restriction by the HUSH-MORC2 corepressor"

Re: Author response to reviewer comments

We express thanks to the editor and all the reviewers for their consideration of our work and their valuable comments, which have substantially improved the manuscript. We have carefully addressed all the comments.

REVIEWER COMMENTS

Reviewer #1 (Remarks to the Author):

This manuscript explores the interplay between DNA methylation and the HUSH complex, and its effector protein MORC2, in mediating retrotransposon repression in a hNPC somatic cell model. The topic is of very high interest. I congratulate the authors on the quality of the CRISPRi experiments and steps taken to measure retrotransposon expression and repression robustly with RNA-seq and CUT&RUN. These are challenging experiments and for the most part they are executed well here. However, I found the manuscript as it stands confusing in parts. I think it would benefit from being more organised conceptually and more considered in its comparisons to the existing HUSH literature, and I feel limited additional experimental data are required to confirm one of the main conclusions.

Thank you for your positive appraisal of the study and its significance. These comments have helped improve the paper substantially. We have reorganized the paper conceptually to provide a clearer and more logical flow, since both reviewers 1 and 2 have made similar points on this. We have also added several analyses and the requested experiments. Please see below for point-by-point responses.

Comments:

1) If I was to summarise the main point (please correct me if I'm wrong) it is that HUSH does not seem to be doing much (Fig. 2) to target L1 in the hNPCs, unless DNMT1 is depleted, which triggers minor L1 transcription leading to HUSH recruitment (dovetailing with the Paul Lehner lab model from reference 30). HUSH depletion in this environment of minor L1 expression then leads to major L1 expression.

Yes, this is an accurate summary of the main point – and we also see the same mechanistic hierarchy playing out at ALRs and protocadherins – the difference being that unlike L1s, these elements are somewhat hypomethylated at steady state in these cells.

In that case, how relevant are the findings from Fig. 2?

This is a fair point, and we see that reviewer 2 also found the logical flow here confusing. Thus, we have reworked the text and figures to improve the clarity of the story and in

doing so moved some of the original Fig 2 analysis to the Supp Fig 3. See below for further discussion of how these data sits in the context of the HUSH literature.

They show MORC2 was at some point recruited to older, potentially 5' truncated L1s lying in sense to gene introns. That seems at odds with the reference 30 model of recruitment to intronless, independently transcribed L1s, yes?

To our understanding our data is not at odds with the Seczynska/Lehner model, which does not require HUSH-targeted L1s to be independently transcribed. Their study mostly used transgenes to elegantly test the dependencies of HUSH recruitment. In the transgene reporters, the promoter is required for transcription and thus, they show, HUSH recruitment (Seczynska/Lehner-Fig2b is illustrative, as is Liu/Wysocka-Nature-ExtDataFig8d). In the Seczynska/Lehner study, analysis of endogenous elements focused on Periphilin RIP-seq experiments in 293T cells (see e.g. Fig 2e; Ext Data Fig 5f). In those snapshots, one can see the same as what we have in hNPCs: the clearest endogenous L1 binding targets of Periphilin are intronic, long or full-length L1PAs, transcribed by upstream genic promoters.

I don't think it is sufficient to explain away this difference based on the methylation landscape of the cells involved either. The hNPCs here have a mean L1 methylation of ~80% and HEK293T and HeLa cells (used in ref 30) have a mean L1 methylation of >75% (ref. 49 Fig. S2). Please consider this reasoning and clarify the significance of Fig. 2 and its relationship to the current model of HUSH silencing.

Thank you for this comment. It is indeed true that - like hNPCs - 293T and HeLa have more methylated genomes than some other cell lines, and this is also true at L1s, as demonstrated by the referenced Cristofari lab study. This is notably different to the K562 cells used in the Liu/Wysocka HUSH-L1 study, where L1s are essentially completely demethylated. In keeping with our model, really drastic endogenous (human) L1 transcriptional upregulation upon HUSH depletion is restricted to K562 cells in the Liu/Wysocka study. To our knowledge there has been no RNA-seq reported on HUSH-depleted 293T cells, but at least two studies report transcriptomic changes in HeLa cells with TASOR knockdowns – Tunbak/Rowe 2020 and Matkovic/Margottin-Goguet 2022. In both studies, the effect on endogenous L1 transcription (re-analysed below using Tetranscripts, new Supp. Fig 4B) is modest, as discussed in both the referenced papers. We would predict that a similar result would be obtained in 293T cells. This is in line with what we and others have observed over the years studying HUSH in those models: yes, binding (and H3K9me3 deposition) is strikingly localized to L1s – often intronic elements, as discussed above. And yes, these cell lines are a good system in which to do careful mechanistic analyses with transgene reporters. But the actual transcriptional consequences on endogenous L1s of removing HUSH, are limited in cells where DNA methylation is high. The exception to this is the (hypomethylated) K562 line. In K562 cells this notably correlates with MORC2 binding to young FL-L1s, as shown by analysis of ChIP-seq data from that study (new Supp Fig 4C).

Supp Fig 4B, C

B) Effect of TASOR depletion on transcription of young L1 subfamilies in different cell lines (quantified by TE transcripts, presented as log₂-fold change (LFC) +/- standard error). Data are from Tunbak/Rowe et al. 2020, Matkovic/Margottin-Goguet et al. 2022, Liu/Wysocka et al. 2018 (GSE135765; GSE184399; GSE95374). **C)** MORC2 binding to evolutionarily young FL-L1s in the demethylated K562 line, assessed by ChIP-seq in Ctrl and MORC2-KO cells. Data from Liu/Wysocka et al. 2018 (GSE95374).

2) I appreciate the authors cite reference 59 in the Discussion but I feel this is inadequate and it should be covered in the Introduction. This prior work from Helen Rowe's lab showed that the impact of HUSH depletion on retrotransposon repression was methylation dependent (mESCs tested in 2i vs serum). This affirms one of the main conclusions of the present work.

Agreed - we have now included reference to this work in the introduction.

3) lines 81 and 389 - cancer cell lines may have aberrant methylomes but, to be fair to those models, the hNPCs used here appear very clonal in their methylomes too (based on the plots in Fig. 3G). Their relevant levels of genome-wide methylation arguably resemble those of the workhorse cancer cell lines used to study HUSH thus far. Can the authors point to methylomes from fetal brain tissue to show the hNPC cell line faithfully recapitulates both the level and heterogeneity of methylation observed throughout the genome? If not, it may not be the case that the hNPC methylome explains differences to HUSH data obtained from cancer cell lines.

The key strengths of the hNPC line we have used are its reproducibility and its unusual tolerance amongst human cells towards DNMT1 removal (expanded on in PMID 31320637). These two advantages have enabled us to query the interplay of DNAm and HUSH in somatic, non-transformed cells – the central aim of this paper. As with all models, however, this one is not perfect. Although not clonal in origin, the line is derived from a small number of cells dissected from fetal brain tissue that have the characteristics of neuroepithelial-like stem cells. It cannot capture the level and heterogeneity of methylation in fetal brain tissue. We are currently working on obtaining high-quality methylomes of fetal brain tissue but feel that extensive comparisons in this direction would

confuse the key points of the paper, and fall outside the scope of this mechanistic study. We have now highlighted these limitations in the first Results section:

"The hNPCs cannot fully describe the epigenetic status or heterogeneity in the developing human brain – and are isolated from a timepoint that slightly precedes the tissue samples analysed by snRNA-seq and CUT&RUN. Nonetheless, these experiments suggest that hNPCs represent a useful model for studying epigenetic mechanisms of repeat regulation in human brain development and confirm that the DNA methylation patterns of somatic human cells are present in the hNPC model."

Regarding the comparison between global methylation levels of our hNPCs and HeLa/293T/K562, we agree and our thinking here is developed above. Indeed, as one might predict from our model, there are not such striking differences in transcriptional changes between HUSH data from (highly methylated) cancer cell lines and hNPCs – only from (demethylated) K562 cells. These are correlations but do largely support our model. We have altered the text in various places to be more specific in our reference to the HUSH literature for different lines (e.g. to highlight the unusual methylation status of K562).

4) The title for Fig. 1 is at odds with what is presented, there is no data for retrotransposons in the figure that I can tell.

We have now included analysis of retrotransposon classes in (new) Fig 1C shown below.

5) L1Hs methylation is high genome-wide in the hNPCs, but do the authors observe the "escapee" L1 loci? As observed elsewhere, these should have unusually low methylation in the hNPCs (PMID: 30692270, ref 2 and ref 38). Whether these full-length L1s have a YY1 site or not (line 228) they can be transcribed (and retrotranspose) and therefore should attract MORC2 and be marked by H3K4me3 in the control hNPCs. Some of these escapee loci are listed in the supplemental of reference 2. The violin plots in Fig. 3B suggest there are a reasonable number of such escapee L1s in the control hNPCs (and they could be identified from the methylartist segmeth output of the ONT data processing step mentioned on line 692).

Interesting question. To investigate this, we downloaded the supplemental table of PMID 31230816, lifted coordinates of the reference elements to hg38 and then carefully examined H3K4me3 one by one over these 27 elements in our datasets: two bulk fetal forebrain tissue samples, two NeuN+ sorted post-mortem adult forebrain tissue samples,

and control hNPCs. To summarize the findings: 1/27 was marked by H3K4me3 in both fetal samples (*chr17:3176531-3182557*) and another 1/27 was marked by H3K4me3 in one but not the other fetal sample (*chr11:93420986-93427031*); 0/27 were marked by H3K4me3 in the adult neurons; 0/27 elements were marked by H3K4me3 in our control hNPCs. Even though H3K4me3 improves mappability of the youngest L1 elements, we considered that mappability might remain an issue due to local genomic context of these insertions. To control for this, we plotted signal in the DNMT1-KO hNPCs and saw that many of the elements gained H3K4me3 upon loss of DNMT1. See the figure below. In all cases the signal was filtered by MAPQ > 10.

Figure rev 1. Assessment of H3K4me3 enrichment over 'escapee' L1s in neural cells and tissues.

Interestingly, we found that one of the hot elements (an intronic insertion in TTC28) was weakly transcriptionally active in hNPCs at steady state despite no detectable promoter H3K4me3. The promoter is largely methylated but we did identify one read with unmethylated CpGs (see below), so it could be that the H3K4me3 experiment was not sufficiently sensitive to pick up activity of this or other hot L1 promoters – or that this element is driven by an upstream promoter. Regardless, this element is indeed regulated by MORC2 (CRISPRi hNPCs DEseq2 with unique mapping: LFC 2.43; padj 1.3e-06 relative to non-targeting controls).

Figure rev 2. DNA methylation data for a 'hot' L1HS element in an intron of TTC28.

We also considered polymorphic L1s that we could identify via our Nanopore data. We ran the 'transposons from long DNA reads' (TLDR) pipeline and identified 53 polymorphic FL-L1s absent in hg38 but present in the hNPC genome. We confirmed that these elements are highly methylated as found for the reference elements. We then plotted our new requested H3K4me3 CUT&RUN data (see also below) from Control, DNMT1-CRISPRi and MORC2/DNMT1-CRISPRi hNPCs, centred on the target-site duplications identified by TLDR, leveraging the unique signal that would spill onto the surrounding reference genome up- or down-stream of the TSD (depending on which strand the L1 is found on). Strikingly, we could clearly see the same mechanistic hierarchy playing out over polymorphic elements: silent in control cells, derepressed in the DNMT1-CRISPRi cells and further expressed in MORC2/DNMT1-CRISPRi cells. We show a couple of examples of the signal over a custom-built genome including the homozygous polymorphic elements.

6) line 202 - the mappability of the youngest L1s (L1Hs) with RNA-seq is arguably overstated here. Other works (summarised in PMID: 32576954) have suggested transcription of these loci is very difficult to measure with short-read RNA-seq. This is one of the reasons why I commend the authors on the use of H3K4me3 in their study but also request they perform H3K4me3 CUT&RUN on MORC2/DNMT1 CRISPRi cells. These data are important to corroborate the analyses presented in Fig. 4, which is the heart of the study, and add to the profiles in Fig. 4G.

Thank you for the suggestion. To bolster our conclusions we have added both long read Nanopore cDNA-seq of Control, TASOR-CRISPRi, MORC2-CRISPRi and DNMT1-KO hNPCs. We have also added the requested H3K4me3 CUT&RUN experiment in MORC2/DNMT1 CRISPRi cells. Both datasets confirm the trends we observed in our original submission. Firstly the ONT-cDNA sequencing illustrates young L1 derepression in the DNMT1-KO but not TASOR or MORC2 CRISPRi hNPCs. Notably, the strand-switching step during Nanopore cDNA library preparation enriches for the 5' end of the elements, and many of the observed reads can be traced to the derepressed antisense promoter.

Secondly, the requested H3K4me3 analysis in Control, DNMT1-CRISPRi and MORC2/DNMT1-CRISPRi cells, here plotted over reference FL-L1s from the youngest four subfamilies.

7) An observation the authors may wish to track down: both of the odd L1s in Fig. 3G are marked by ENCODE as enhancers (enhD). Perhaps their low methylation is a reflection of a regulatory role?

Thank you for this insight, we have now noted this possibility in the text.

8) Why is the MORC2 binding so weak in the examples provided in Fig. 4G compared to Fig. 2H? If HUSH is being recruited to these sites by the active transcription of the L1 promoter presumably one would observe more MORC2? (another reason to do H3K4me3 on the MORC2/DNMT1 CRISPRi cells, to confirm these as sites).

Thanks for bringing up this interesting point that we have thought about a lot. To the extent that CUT&RUN peaks are a quantitative readout (a debatable but probably reasonable prior), we can offer two possibilities:

1: Low mappability of the youngest elements. This is probably part of the picture, but allowing for multi-mappers, the signal only increases a little, so we don't think that's fully satisfactory.

2: The amplitude of transcription happening through the element. It is important to keep in mind that the older, genic L1s are being transcribed as part of highly expressed gene transcripts. Thus, the sheer amplitude of transcription – a key determinant of HUSH, and thus MORC2, recruitment – is at least an order of magnitude greater at these sequences than the over the youngest L1s activated by DNMT1 loss. One can observe a similar trend at the clustered protocadherin locus. The signal for MORC2 over *TAF7* (a very highly transcribed single exon gene that divides the beta and gamma *PCDH* clusters) is much more striking than over the neighbouring (more lowly-expressed) protocadherin genes. See Fig 5F inset below.

9) line 407: this sentence could use a qualifier as at present the evidence presented does not demonstrate independence of DNA methylation and HUSH-MORC2 restriction.

Thank you for this comment. We have now avoided the word independence throughout the manuscript since it can be interpreted in different ways.

10) It would be helpful to show some full-length L1Hs loci from Fig. 3E in the style of the ONT and CUT&RUN profiles from Fig. 3G. If these can indeed be mapped uniquely with

RNA-seq then presumably clear examples in support of the genome-wide trends shown in Fig. 3D will be found?

We have now included two examples in a new Fig 2I, which now include short- (Illumina) and long-read (Nanopore) cDNA-seq.

11) For the ONT analysis, as shown in Fig. 3B, where were the coordinates of L1Hs copies obtained from, and did these cover only 5'UTR regions or the entire L1Hs (which tends to underestimate changes due to L1 body methylation being resistant to change).

We have now analysed both FL-L1HS elements and the 5' UTR separately and present violin plots accordingly in (newly numbered) Fig 2B. In a new Supp Fig 2, we plot the methylatin landscape over all reference FL-L1HS elements. We observe high methylation levels across nearly all elements.

Fig 1B

Supp Fig 2.

Note: I previously commented on a preprint version of this work on social media, which I see has been acknowledged in the manuscript. I don't see this as an issue in providing a formal review but readers may wish to be aware of it.

Geoff Faulkner (University of Queensland)

Reviewer #2 (Remarks to the Author):

In this study, Pandiloski et al address a critical knowledge gap related to the interplay between the HUSH complex and DNA methylation in the regulation of genomic repeats, specifically LINE-1 retrotransposons. While it is known that HUSH binds LINE-1 transcripts and recruits SETDB1 and MORC2 to repress chromatin, the specific relationship between HUSH and DNA methylation - a central regulator of repeat transcription - has remained largely unexplored.

Previous studies into the role of the HUSH complex in silencing retroelements often used models with low levels of DNA methylation. To overcome this limitation, the researchers turned to human neural progenitor cells (hNPCs), which exhibit high levels of DNA methylation but can tolerate the depletion of DNMT1 and therefore study the role of the HUSH complex in both scenarios within the same system using CRISPRi. This approach was successfully executed in this study.

Their key conclusions are that even in the absence of MORC2 or the HUSH subunit TASOR in hNPCs, LINE-1 retrotransposons remain silenced due to robust promoter methylation. However, when genome demethylation occurs upon depletion of DNMT1, and evolutionarily young LINE-1 elements become active, MORC2 binding, and presumably HUSH component binding increases to these regions. They suggest that silencing is therefore layered and that simultaneous depletion of DNMT1 and MORC2 results in a substantial accumulation of LINE-1 transcripts (higher than in the single depletions).

This observation aligns with previous research conducted in models such as mouse embryonic stem cells and cancer cell lines, which have low levels of DNA methylation. In these models, deletion of the HUSH complex leads to increased LINE-1 expression and a type I interferon response. The researchers also extend their study to pericentromeric α -satellites and clustered protocadherin genes, repetitive elements that play important roles in chromosome structure and neurodevelopment, with the latter being previously reported during mouse brain development and in brain organoids.

This paper is important and sheds light on the critical functions of HUSH and MORC2 in the context of low DNA methylation levels in development. It contributes to the growing body of knowledge regarding the intricate mechanisms governing epigenetic control during development and in somatic cells. However, as discussed below I did have concerns with their model and would recommend publication if these can be addressed.

Thank you for your positive overall appraisal of the study and its significance. This review has helped improve the paper substantially. We have reorganized the paper conceptually to provide a clearer and more logical flow, since both this review and reviewer 2 have made similar points on this. We have also added several analyses and the requested experiments. Please see below for point-by-point responses.

Major issue

Their general model is that there are two independent layers of regulation and that DNA demethylation via loss of DNMT1 sensitizes L1 elements to HUSH-MORC2 restriction (Figure 4H). However – for the reasons stated below this does not completely make sense to me.

158 Following MORC2 and TASOR depletion experiments they observed loss of H3K9me3 over hundreds of sites, confirming the HUSH pathway as functional in their hNPC model. Furthermore, cut and run experiments showed MORC2 bound to the majority of these sites. This is what would be expected for HUSH-dependent loci. However – it is then difficult to reconcile these results with the next section of their manuscript: On line 210 they don't see a major transcriptional deregulation of L1s in hNPCs lacking MORC2 or TASOR – nor an effect on K4 but they don't report the effect on K9. They find that the loss of DNMT1 does not affect H3K9me3 deposition (Supp Fig 3C), which presumably implies that there was K9 at these loci – but they haven't shown that it is MORC2/TASOR dependent (or not).

Yes, young FL-L1s (by which we mean those from L1HS-L1PA4 subfamilies, which can be independently transcribed) are indeed covered with H3K9me3 in control cells and this is not dependent on TASOR or MORC2 in hNPCs. (The HUSH-dependent L1s at steady-state are older elements in transcribed genes, which are passively transcribed.). We have now shown this in Fig 2F.

F H3K9me3 profiling of individual FL-L1s

Fig 2F.

This would seem to be essential as the basis of their subsequent model relies on the fact that in the next section: 'DNA methylation status controls L1 sensitivity to HUSH-MORC2 restriction' they explain that 'loss of DNA methylation activates transcription of young FL-L1s, which become targets of MORC2 (Fig. 4G,H).' At first glance this seems to make sense – but if prior to the release of DNMT1 they were not already targets of MORC2/TASOR (which requires active transcription) – how did those loci acquire K9 - if they were not transcriptionally active i.e. it seems to become a 'chicken and egg' situation – that requires an explanation – nor are the two 'independent' as they seem to suggest?

This is an excellent point which we understand as addressing two key questions for the field: (i) how is H3K9me3 established over young FL-L1s in early (human) development? and (ii) how is H3K9me3 maintained over these elements in somatic cells? Neither of these questions are the central focus of our study, but to the second point, we show above that young FL-L1s are largely ignored by HUSH-MORC2 at steady-state in hNPCs. To the first point, the molecular basis for how H3K9me3 is established in early development remains an interesting and active topic of research. Various studies have described L1 transcription in early development during the period of hypomethylation, and our model would predict that in those settings, HUSH would indeed bind to the expressed elements (and perhaps be necessary for H3K9me3 establishment). One study which supports our conclusions (albeit in a transformed cell line) is the HUSH-L1 study from Liu/Wysocka. Much of the work in that paper was done in K562 cells, where L1s are completely demethylated (see also Reviewer 1 point 1). Indeed in this context, MORC2 is clearly bound to most young FL-L1s. We have now added this analysis to a new SuppFig4C.

MORC2 ChIP-seq in K562 cells over young FL-L1s
(Liu et al.)

Supp Fig 4C

To expand on this further:

The section:

Line 255 DNA methylation status controls L1 sensitivity to HUSH-MORC2 restriction is important but is written and presented in a confusing manner –it would benefit the reader if they were able to present a clearer flow of events

We apologise for the confusion. Since reviewer 1 also requested a similar change, we have now substantially reworked this and the previous section to try to clarify the logical build up of our model. We hope this will help readers and thank the reviewer for bringing this to our attention.

Line 265: Firstly, they don't interpret Fig 4A – and simply say: 'Upon depletion of DNMT1 we could measure accumulation of MORC2 over activated, young FL-L1s (Fig. 4A).' I'm not sure how to interpret the right-hand panels of Fig 4A? and – what would happen to MORC2 in the absence of TASOR? As discussed above, It's unclear as to how much MORC2/TASOR/K9 is already present at these L1 elements in the presence of DNMT1 (ie at steady state) – as more appears to be recruited in the absence of DNMT1? This is important as the reader needs to know whether these are loci which would (i) be normally classified as 'HUSH-dependent loci' – or (ii) do they 'become HUSH dependent following the loss of DNA methylation', as they become transcriptionally active and recruit HUSH/MORC2 components?

We think it's the second i.e. young FL-L1s are not HUSH dependent elements at steady-state but become HUSH-dependent upon loss of DNA methylation. See the model in Fig 4H.

In this cellular model, essentially all of these young FL-L1s are silent. Their promoters are methylated and the bodies of the elements are H3K9me3-modified as described above. Neither is HUSH/MORC2 dependent according to our data, nor is MORC2 bound to these elements at steady-state. Upon DNA demethylation and concomitant transcriptional activation, MORC2 accumulates over the newly activated young FL-L1s. We have now provided a fuller explanation:

'In the MORC2 CUT&RUN experiment, we found little evidence of MORC2 binding to the evolutionarily youngest FL-L1s in control hNPCs, consistent with the model that these elements are largely ignored at steady-state when silenced by DNA methylation. However, upon depletion of DNMT1 and concomitant genome demethylation, we could observe clear accumulation of MORC2 over the 5' end of those youngest FL-L1s (Fig. 3C).'

One would presume it is in fact the former – as they should have been 'HUSH-defined' by the presence of TASOR/MORC2-dependent deposition of K9. However, if this is indeed the case, how is the K9 deposited if HUSH components are not present and they are not transcriptionally active? This all needs to be clearly explained to help understand their model.

Again, we apologise for the confusion. The 'HUSH-defined' L1s at steady state are those passively expressed from upstream promoters, typically older L1s. See also the comments

from reviewer 1. We have now moved that H3K9me3 analysis to the Supplemental (Fig S3) – which, though interesting may confuse the flow of the story.

For example in Fig 4G – there is increased gene expression following depletion of DNMT1, with MORC2 recruited. Furthermore, derepression becomes most marked following the double depletion. But for these genes they don't show the levels of K9 deposition. This is confusing as either (i) there is no K9 present in the control – in which case how could they initially define this gene as 'HUSH-dependent' or (ii) there is K9 deposited – which is also confusing – as how does it get deposited in the absence of transcription and in the absence of MORC2/TASOR recruitment to that locus?

In this paper we have mainly looked at all young FL-L1s (L1HS-L1PA4) because these elements have the capacity to be independently transcribed and are controlled by DNA methylation. Regarding the specific question regarding H3K9me3 deposition – there is H3K9me3 over these elements in control cells and this is usually not dependent on TASOR or MORC2, as we discuss to a similar point above. H3K9me3 maintenance at these silenced elements in this cell type thus appears not to depend on transcription. Whether the establishment of H3K9me3 at these elements in early development depends on HUSH-MORC2 is certainly a possibility.

In fact the title of the previous paragraph section (DNA methylation, but not HUSH-MORC2, controls L1 transcription in hNPCs) is misleading as it gives the strong impression that L1 silencing is maintained by promoter DNA-methylation and that in the absence of MORC2/TASOR, at least for the majority of L1 elements, silencing is maintained i.e. that MORC2/TASOR are irrelevant for silencing these L1 elements (yet they have K9?)

Yes, this is the case and is an important observation: young FL-L1s are maintained in a silent state in hNPCs even upon loss of HUSH-MORC2. These have H3K9me3 but as discussed above, the maintenance of this methylation is not HUSH-MORC2 dependent (nor is MORC2 bound).

The fig legend for Fig 3 reads: 'FL-L1s are generally silenced by DNA methylation but not HUSH-MORC2 in hNPCs'. However, in the following paragraph they go on to show that the effect is clearly much more nuanced – that in fact there is a layering effect, and that there is only maximal release of L1 element repression following the double depletion – i.e. DNMT1 and MORC2. This latter point makes sense and is logical, so it makes the first title incorrect? Both the title and the content of the previous paragraph (DNA methylation, but not HUSH-MORC2, controls L1 transcription in hNPCs) needs rephrasing – so as not to confuse the reader that L1 silencing is all about DNA promoter methylation?

They subsequently contradict their previous title by summarizing their work: 285 'Simultaneous loss of DNMT1 and MORC2 therefore causes a massive accumulation in the levels of young FL-L1 transcripts, suggesting independent transcriptional and co/post-transcriptional L1 control by the two epigenetic pathways.'

We have now substantially reworked this and the previous section in an attempt to provide more logical flow. Since the word 'independent' could be interpreted in different ways

(and has been by the three reviewers), for clarity we have now avoided this word throughout the paper.

ALRs

They could take the opportunity to emphasise that, presumably because the ALRs are hypomethylated, the effect of MORC2/TASOR depletion is more marked here than in other regions of NPCs and in fact shows a more major effect than the DNMT1 deletion? – in fact they don't really mention the effect of the loss of MORC2/TASOR here at all?

Thank you for bringing this up since this is the point we are trying to make. We note that in the original submission we wrote:

'In subfamily-level analysis of RNA-seq data upon loss of MORC2 or TASOR individually, the alpha-satellite-like repeat (ALR) was by far the most upregulated class (Fig. 5A).'

We have now added a statement to spell out that indeed we think this is because they are somewhat hypomethylated at steady-state:

'We therefore considered the DNA methylation status of ALRs. Although ultra-long (>100-kb) reads are required to uniquely map higher order arrays of ALRs at centromeres,⁴³ the more scattered distribution of ALRs in pericentromeric regions allowed for unique mapping of thousands of our Nanopore reads (N50, 14-kb) that span relatively short ALR arrays and/or contain unique flanking sequences to aid mapping. We found that CpGs in pericentromeric regions containing ALRs were around 50% methylated, significantly below the whole genome average (Fig. 5B,C). Thus, ALRs are somewhat hypomethylated and subject to HUSH-MORC2 restriction at steady-state in hNPCs.'

Their statement L325 that: DNA methylation status was unchanged over ALRs in cells depleted of MORC2 – suggesting that the pathways operate independently, as was observed at L1s is not necessarily correct. If DNA methylation machinery was recruited by HUSH components (as is certainly possible) – then the loss of MORC2 would indeed prevent silencing, but HUSH would still likely be recruited i.e. DNA methylation might still be dependent on core HUSH components – if not directly dependent on MORC2 – so the two would then be interdependent and not independent?

This is an interesting point and indeed it could be the case the DNA methylation could depend on other HUSH components (this could also depend on cell type and developmental stage). Regarding the term independent, this and other comments above and from other reviewers raise a common theme that the word can be interpreted in multiple different ways. We have therefore now avoided this term throughout the revised manuscript.

General impression is that the ALRs are more HUSH sensitive because they are hypomethylated – why are they hypomethylated?

This is another interesting point that we reflect on briefly in the discussion. It could be because ALRs are rather AT-rich, but very little is known mechanistically about ALR epigenetics owing to the challenges of studying them mechanistically with most technologies. We note that it is a known phenomenon that ALRs are hypomethylated in ageing and various pathological states (reviewed in PMID: 34082816) but the drivers of these changes are not known.

Other points:

Introduction

DNA methylation was examined in the context of an SVA reporter by Robbez-Masson, L et al 2019 and its recruitment was HUSH dependent so this paper should be cited early
We have now cited this paper in the introduction.

67 The human silencing hub (HUSH) complex²¹ has emerged as an important epigenetic
68 regulator of repeats through H3K9me₃.^{7,22–24} - The correct references here should be
21 – 24.

Amended.

Line 70

Recent data have shown that HUSH also participates as an adapter for cotranscriptional RNA processing complexes, leading to the destruction or termination of targeted transcripts.^{26–28} Such an RNA-dependent mechanism is reminiscent of evolutionarily ancient systems such as the yeast RNA-induced transcriptional silencing⁷³ (RITS) complex, and indeed analysis of HUSH protein architectures revealed striking similarities with RITS.²²

The combination of these two sentences doesn't seem logical – it reads like a non-sequitur. The first relates to the link between HUSH and the destruction/termination of transcripts (NEXT complex). The second relates to the silencing mechanism of HUSH which is indeed evolutionarily related to RITS, as both are RNA-dependent silencing. The link between the two is not about how HUSH relates to the degradation machinery?

Thank you for spotting this. We have now changed the order of the sentences to restore the logic.

'Recent data have shown that HUSH also participates as an adapter for co-transcriptional RNA processing complexes, leading to the destruction or termination of targeted transcripts.^{26–28} Endogenous genomic repeats targeted by the HUSH-MORC2 co-repressor – that is, L1s and certain repetitive genes^{7,21,25,29} – appear to be unified by the presence of long, intronless transcriptional units.³⁰ Experiments with transgene reporters have shown that transcription is required for recruitment of the complex to the reporter.^{7,27,30} A transcription-dependent mechanism of H3K9me₃ deposition is reminiscent of evolutionarily-ancient systems such as the yeast RNA-induced transcriptional silencing (RITS) complex, and indeed analysis of HUSH protein architectures revealed striking similarities with RITS.²¹

Results

Line 167 – should note reference 30

Amended.

How representative is the NPC cell line – was it clonal in origin – more details would be helpful here?

Although not clonal in origin, the line is derived from a small number of cells dissected from fetal brain tissue that have the characteristics of neuroepithelial-like stem cells. It thus cannot capture the level and heterogeneity of methylation in fetal brain tissue, and we have added a statement to that effect in the first Results section and the Methods section. The description of the derivation of the cell line is detailed in the study from Anna Falk and Austin Smith, cited in both Results and Methods sections (PMID: 23884946). We have briefly expanded on this in the Methods section:

'The embryo-derived human neural progenitor cell line Sai2, derived from a small number of cells dissected from fetal brain tissue at Carnegie stage 16 (approximately 6w post-conception), and which has the characteristics of neuroepithelial-like stem cells,38 was cultured according to standard protocol.70'

Several additional elements could be included to further validate the system; it would be helpful to incorporate other markers that characterise the identity of the cell line, such as Sox1, Sox2, Musashi-1. Moreover, protein expression of the HUSH complex (by western blotting or immunofluorescence) would be a helpful addition to the 2019 proteomic data. We have now included a heatmap showing expression of a number of marker genes (including SOX1, SOX2 and MSI1 – and several others) in Supp. Fig 1H. We have also included a staining of SOX2 protein in Supp. Fig 1B (below). Western blots illustrating TASOR (the defining HUSH component) and MORC2 expression are shown in Fig 1J.

When presenting the genome-wide DNA methylation plot in Figure 1D, it would be valuable to dissect it in some of the repetitive sequences - this could provide insights into whether these sequences are completely methylated and to study and discuss the evasion of DNA methylation in certain LINE-1 elements as reported previously in different studies.

This is an excellent idea, thank you, we have now dissected this plot to show the whole-genome average and retrotransposons (LINE-1, SVA and HERV-K families, see below).

Fig. 1C

Can they address the temporal discrepancy between the snRNAseq analysis, which covers the period of 7.5-10.5 weeks post-conception, and the characterization of the hNPCs, which are at 6 weeks post-conception. The same is seen with CUT&RUN profiling of H3K4me3 and H3K9me3 in human fetal forebrain samples at 7- and 10-weeks post-conception. Discussing how these factors change during embryonic development would provide a more comprehensive understanding of the research and clarify these points.

This is an important point. We have now added text to this effect in the first Results section:

The hNPCs do not fully describe the epigenetic status or heterogeneity in the developing human brain, and are isolated from a timepoint that slightly precedes the tissue samples analysed by snRNA-seq and CUT&RUN. Nonetheless, these experiments suggest that hNPCs represent a useful model for studying epigenetic mechanisms of repeat regulation in human brain development and confirm that the DNA methylation patterns of somatic human cells are present in the hNPC model.

The authors emphasise that MES cells and cancer cell lines may show incomplete or aberrant DNA methylation – are the methylation patterns of their hNPCs representative of normal somatic cells? How does 80% genome wide CpG methylation relate to other cell types?

This is another important point. As was discussed in response to reviewer 1, cancer cells are highly variable in their methylation status and we have now added analysis of published datasets from HeLa and K562 cells which have very different DNA methylation patterns which correlates with the effect on L1 transcription of HUSH removal, strengthening our model. The ~80% methylation is typical of somatic human cells (PMID: 7079182, now referenced).

Do the accumulated L1 transcripts have any impact on cell fitness, neural differentiation capacity, or any phenotypic outcomes? The same question applies to the dysregulation of ALRs and protocadherin genes. For instance, while the upregulation of the IFN response is demonstrated in individual CRISPRi experiments, it is not shown for the double depletion

of DNMT1 and MORC2. Is it also increased in the double depletion compared to the single ones?

The loss of TASOR or MORC2 in hNPCs do not have measurable consequence on cell fitness or identity in our hands, and can be further differentiated from hNPCs to neurons despite deregulation of protocadherins and ALRs. The phenotypic consequences were extensively characterized *in vivo* in mice with a conditional nervous system KO of MORC2a or MPP8, which led to enlarged brain:body ratio correlated with protocadherin deregulation (PMID 36332029, discussed in the paper). With regards to hNPCs lacking DNMT1 (either through acute knockout or CRISPRi), we have found that the cells remain proliferative at least to day 15 post-transduction. However, they divide at a slower rate once DNA methylation is lost (see PMID: 31320637) and attempts to further differentiate these cells to neurons have led to a high level of cell death. We note that we did not see upregulation of the IFN response in TASOR or MORC2 depletions. However, there was upregulation of IFN-stimulated genes in hNPCs lacking DNMT1 which was consistent between DNMT1-KO and DNMT1-CRISPRi treatments, as shown in a Supp Fig 5E. We found that this was not increased in the DNMT1-MORC2 double depletion compared with DNMT1 alone.

Fig 2 – I was a bit confused by the MORC2 locus – as Douse has previously shown that the MORC2 locus is already HUSH repressed – or at least that there is K9 deposited in a HUSH dependent manner? –

Indeed, *TUG1* (a lncRNA adjacent to *MORC2*) is marked by HUSH-dependent H3K9me3 in HeLa cells and here in hNPCs. The reason that we focused on the *MORC2* gene (rather than the whole locus including *TUG1*) was that we thought it could be confusing to readers less expert with the system, since both CRISPRi and HUSH-MORC2 are associated with H3K9me3. The purpose of that figure panel (now Fig 1H) was only to show targeted H3K9me3 accumulation downstream of the TSS upon dCas9-KRAB binding (CRISPRi). However, when one does zoom out a bit one can indeed see both effects at one locus in the CRISPRi cells: accumulation of H3K9me3 at the TSS, and loss of H3K9me3 over *TUG1*. In TASOR CRISPRi cells, of course the entire H3K9me3 peak is lost since there we are only monitoring HUSH activity and not dCas9-KRAB activity. The locus is therefore a neat confirmation of both CRISPRi targeting and a functional consequence of the depletions.

So on Line 149, it might be worth pointing out that the endogenous MORC2 locus, unlike the TASOR locus, is known to be HUSH repressed – as shown in Figure 2B where there is K9 present? However, I was then surprised that in figure 2D they do not see an increase in MORC2 protein expression following TASOR depletion?

Great idea, we have now added a comment to this effect in the text.

'We note that the MORC2 locus is itself HUSH-repressed,²¹ and indeed saw clear evidence for loss of H3K9me3 over the MORC2 promoter in TASOR-CRISPRi hNPCs, correlated with an increase in transcript levels, confirming a functional knockdown'.

We were also surprised that we did not detect an increase in MORC2 expression following TASOR depletion in line with the clear increase in transcription (RNA-seq and promoter H3K4me3). We did repeat the blot a further two times (with cell pellets from separate transductions, see below). We saw evidence of a small increase in MORC2 expression in TASOR CRISPRi cells

Fig 2E – I wasn't clear what information I was supposed to gain from this figure?
 The purpose of Fig 2E in the original submission was solely to validate that hNPC identity was not affected by the CRISPRi treatments i.e. that changes observed in RNA-seq or CUT&RUN were not indirect consequences of a change of cell identity. We have now moved this to Supp Fig 1G and added a brief interpretation to this effect in the legend.

Fig 2F – the legend says: 'Volcano plot showing H3K9me3 changes genome-wide over H3K9me3 peaks in MORC2 (n=2) and TASOR (n=2) CRISPRi hNPCs compared to controls (n=4) as measured by CUT&RUN epigenome profiling 10 days post-transduction (significant points defined by fold-change >3 and Poisson p-value <1e-04). It wasn't clear how this experiment was performed – have they just pooled data from MORC2 and TASOR depletions? Why aren't they showing the differential effect of a MORC2 and a TASOR depletion – are we to assume that in the absence of MORC2 or TASOR the same genes are affected? it would be clearer to show separately and clarify that they behave similarly and how different the double depletions are.

Apologies for the vagueness which was also commented on by Reviewer 3. The dots are H3K9me3 regions defined as peaks in at least two of the samples (control hNPCs n=4, MORC2 CRISPRi hNPCs, n=2, TASOR CRISPRi hNPCs, n=2) compared to their IgG controls. The regions defined as H3K9me3-depleted in CRISPRi were significantly lost in both MORC2 and TASOR CRISPRi treatments i.e. we pooled the MORC2i and TASORi replicates as the treatment group (n=4) to compare with the control group (n=4) in differential enrichment analysis. We did this to ensure we were working with bona fide regions where the HUSH-MORC2 corepressor is bound (not just MORC2 or TASOR), and to avoid off-target effects that may be associated with CRISPR-dCas9 approaches. We also compared the MORC2 and TASOR CRISPRi treatments separately: this is now shown in Supp. Fig. 3A-D and indeed shows a correlation between the two datasets, reproduced below.

Lines 163 – 168 The data shown in Fig 2I is a striking observation and supports the model suggested in ref 30 and should be attributed as such in the text? The authors discuss that HUSH-dependent silencing primarily targets young active L1s retrotransposons located on the same DNA strand as the host genes; however, in multiple figures such as in figure 4G the authors show antisense copies to the host gene. It would be helpful to clarify why these examples were chosen.

Absolutely, it is striking and we now reference ref30, the omission of which was purely a typo that is now amended and for which we apologise. We should point out again here, however, that HUSH-dependent H3K9me3 over these elements (which are typically older and transcribed as part of host genes) is not equivalent to silencing. Since the data around these intronic elements and H3K9me3 has been a key source of confusion in the manuscript we have moved the analysis to the supplementary information to keep the focus on the main message of the paper, which is about interplay of HUSH-MORC2 and DNAm in transcriptional regulation of repeats. Please see also responses to reviewer 1.

Regarding fig 4G, we have simply chosen FL-L1s from the youngest subfamilies to illustrate the model. Here the genomic context is not important and we could've chosen many other examples. We have now changed the examples shown to avoid confusion around this, now showing one (antisense) intragenic element and one intergenic element. We have also now included analysis requested by Reviewer 3 that demonstrates how upon DNA demethylation, MORC2 affects L1 transcription independent of their position/orientation within genes, see new Supp Fig 6D. This is further support of our model.

FL L1 (>6kb)	n	genic	intergenic	Genic sense	genic antisense	%sense
total	6509	3361	3148	1118	2243	33
"HUSH bound" steady-state hNPCs	181	144	37	127	17	88
DNMT1i-upreg	1216	678	538	190	488	28
MORC2-DNMT1i-upreg	2504	1342	1162	421	921	31

D

Genomic context of targeted FL-L1s

-Line 160 they show MORC2 bound to the majority of these sites – can they be more quantitative than this.

We have now included quantitative analysis: using HOMER differential enrichment analysis we confirmed direct MORC2 binding at around half (262/532) of targeted sites ($p < 0.0001$). The heatmap shown in Supp Fig 2I (reproduced below) suggests more than this are occupied by MORC2 but these regions fall just below the strict cutoff.

I

-Doesn't Fig 4D need a control CRISPRi to see if there is any effect of TASOR/MORC2 depletion and can they show a statistical analysis of these results? Maybe their data is already analysed relative to the control – in which case this should be clearly stated.

Apologies that this was not clear. In all cases the log fold-change reported in the bar charts are relative to a CRISPRi control in which we transduced with the same vector expressing both the dCas9-KRAB transgene and a gRNA to a LacZ sequence not present in the human genome. We have double checked the labels and legends of this and other relevant panels to clarify that the changes are relative to the control.

-Was there anything significant about the 19 FL-L1s which were upregulated with HUSH loss – do they know why these loci were aberrantly DNA-methylated? The significance of this statement is somewhat unclear:

Interestingly, upon examination of the promoters of the three candidate elements in our Nanopore sequencing dataset we could detect a proportion of demethylated reads (Fig 3G and Supp. Fig. 3E), suggesting that these elements are incompletely methylated in hNPCs.

I assume they are implying that these sites weren't methylated and therefore needed MORC2/TASOR for repression – but it would be helpful to know if this was unique to these sites – are they suggesting that all the young L1s which were not HUSH-sensitive are fully methylated?

Yes, the vast majority of young FL-L1s appear to be fully methylated in this hNPC model. To illustrate this we plotted the methylation of individual FL-L1HS copies where you can clearly see only a handful of elements that have partial demethylation of the 5'UTRs. This is now presented in Supp Fig 2.

As for why these candidate elements are demethylated, we don't know. However, Reviewer 1 pointed out that these three candidates are annotated by ENCODE as distal enhancers. We have now noted this in the text.

'Interestingly, examination of Nanopore sequencing reads over the three candidate elements revealed a proportion of demethylated Nanopore reads in each case (Fig. 3A). We note that all of these elements annotated as distal enhancers by ENCODE, which may explain their partial demethylation.'

Discussion

413 Why other repeat transcripts (e.g. SVA retrotransposons and certain LTR elements, activated in cells lacking DNMT1) remain ignored by HUSH MORC2 remains an open question.

Actually – It's not such an open question – they don't fulfill the criteria for HUSH-dependent repression

Thank you for bringing up this point that we largely agree with. So far the criteria for HUSH-dependent repression could be broadly summarised as 'A-rich, >2kb, transcribed'. However, there are elements that do not fulfil this criteria. We think it is fair to say – especially given that these are rather recent findings – that more investigation is required. SVAs are CG-rich, yes, but many are >2kb. Conversely, most LTR elements are truncated fossils or solo LTRs that are not transcribed. But some LTR elements do fulfil the criteria. As a compromise we have removed the phrase 'open question' and now instead write:

Why other repeat transcripts (e.g. SVA retrotransposons and certain LTR elements, activated in cells lacking DNMT1) remain ignored by HUSH-MORC2 requires further investigation but may be related to the CG-rich sequence composition of those elements. Details of ALR transcript structures and promoters remain largely uncharacterized, but two features of ALRs: an A-rich sequence and long transcriptional units, now seem to be characteristic of HUSH-sensitive elements 30.

We hope that phrasing satisfies the reviewer.

Reviewer #3 (Remarks to the Author):

In the manuscript entitled "DNA methylation governs the sensitivity of repeats to restriction by the HUSH-MORC2 corepressor", Pandiloski et al explore the role of the HUSH complex in silencing repeat sequences (primarily L1s) in human neural progenitor cells and its interplay with DNA methylation machinery. This is relevant because, as the authors note, the majority of studies that have explored L1 regulation by HUSH have been done in cells (K562, ESCs) where DNA methylation is strongly reduced, compared with most somatic cell populations where CpG methylation is much higher. Using an elegant CRISPRi approach against TASOR (a core HUSH component) and MORC2 (a peripheral HUSH component and ATPase) the authors show that a subset of L1 elements are targeted by HUSH to establish H3K9me3. These elements share some unique features, they are enriched in genes, enriched in the sense orientation, and display low levels of transcription and hypomethylation. L1 elements are not broadly activated however upon MORC2 or TASOR loss, unlike upon Dnmt1 loss, where massive L1 activation and promoter H3K4me3 appearance are observed. Interestingly, upon Dnmt1 loss, MORC2 is more strongly recruited to young L1s, and this recruitment is important for reducing the accumulation of L1 transcripts, as there is a synergistic effect on L1 levels upon Dnmt1 and TASOR/MORC2 loss. Finally, the authors observe that pericentromeric ALR repeats, and members of the PCDH gene family, are regulated by HUSH, and like L1s share some similar features, namely they are somewhat hypomethylated relative to most of the genome, and are regulated by a combination of DNA methylation and HUSH.

The manuscript is generally well written and likely to be of broad interest across fields of evolution, neurobiology, genomics, and chromatin biology. The authors have developed a useful model to query the interplay between global DNA methylation and sensitivity to HUSH-MORC2 repression and the presented data adds to the mechanistic understanding of HUSH complex mediated regulatory control of endogenous repeats. This study adds to the growing body of evidence that the HUSH complex functions to dampen the expression of expressed repeat sequences and clarifies the relationship between DNA methylation and HUSH mediated control of L1 elements.

Thank you for this clear summary and positive overall appraisal of our study and its significance. This review has helped improve the paper substantially. Please see below for point-by-point responses.

Some comments for the authors consideration:

Figure 2F: It is unclear what the dots represent, are these peak regions? Individual repeats? Small windows of the genome? I get that they represent areas where H3K9me3 is either changed or unchanged, but it is unclear how they were defined. Furthermore, it is not immediately clear how the data shown in Figure 2F was generated as the description of "pooling replicates" in the results (line 157-158) is a bit vague. A brief explanation in the results would help the reader follow the criteria used to identify the 532 methylated regions identified as lost in CRISPRi. You could consider adding a panel to Figure 2 or supplemental – for example is the fold change correlated between the two KD datasets for the 532 methylated regions identified as lost in both?

Apologies for the vagueness. The dots are H3K9me3 regions defined as peaks in at least two of the samples (control hNPCs n=4, MORC2 CRISPRi hNPCs, n=2, TASOR CRISPRi hNPCs, n=2) compared to their IgG controls. The regions defined as H3K9me3-depleted in CRISPRi were significantly lost in both MORC2 and TASOR CRISPRi treatments i.e. we pooled the MORC2i and TASORi replicates as the treatment group (n=4) to compare with the control group (n=4) in differential enrichment analysis. We did this to ensure we were working with bona fide regions where the HUSH-MORC2 corepressor is active (not just MORC2 or TASOR), and to avoid off-target effects that may be associated with CRISPR-dCas9 approaches. We also compared the MORC2 and TASOR CRISPRi treatments separately: this is now shown in Supp. Fig. 3A-D and indeed shows a correlation between the two datasets, reproduced below.

Figure 2G: As LINE elements are more abundant than LTRs in the genome, it may be most appropriate to provide information on relative enrichment rather than absolute numbers.

This is an important point. How best to do this is not as trivial as it first sounds because H3K9me3 is highly enriched over retrotransposons at baseline (and there is likely a length dependence to this too). We have thus repeated this analysis in a simple way by doing the following: we counted the number of transposon overlaps with the 532 HUSH-MORC2-dependent H3K9me3 peaks. We then took 532 random H3K9me3 peaks and made the same counts, and compared these numbers. The L1 enrichment was significant, as shown in new Supp Fig 2B (see above).

Figure 2I: Is expression of the genes flanking the 181 HUSH-targets altered? (is this the 139 HUSH-bound genes shown in Supplemental 3A? if so why 181 vs 139?)

Yes, this was the analysis shown in Supp3A (now Supp3J). To clarify the numbers, some of the 181 FL-L1s are classified as intergenic, leaving 144 elements overlapping genes. A handful of these are multiple elements in the same gene, leaving 139 genes.

Figure 4D-F: My assumption is that the L1s affected by DNMT1/MORC2 loss do not show the same bias as those shown in figure 2. Can the authors make a figure that shows that upon DNA methylation, MORC2 affects L1s independent of their position/orientation within genes?

Thank you for this smart suggestion. The assumption is correct: the L1s deregulated by DNMT1 CRISPRi and DNMT1/MORC2 double CRISPRi treatments show no such bias on position or orientation. We have included this analysis as Supp. Fig 5D.

FL L1 (>6kb)	n	genic	intergenic	Genic sense	genic antisense	%sense
total	6509	3361	3148	1118	2243	33
"HUSH bound" steady-state hNPCs	181	144	37	127	17	88
DNMT1i-upreg	1216	678	538	190	488	28
MORC2-DNMT1i-upreg	2504	1342	1162	421	921	31

D

Figure 5: I suggest providing methylation data for the protocadherin locus as in Figure 3 (presuming this data is readily available in the existing nanopore dataset).

Yes, this data is shown in Supp Fig. 9D

Figure 5F. In this zoomed out view, it appears MORC2 may have a few binding sites within and near the PCDHgamma cluster. Can the authors show zoomed up views at these peaks?

Are these peaks overlapping L1s? The authors mention that MORC2's strongest peak is at the TAF7 gene but this is not labeled. Do they authors have a model for how MORC2 is recruited here (beyond that it is a single exon gene)? Is this a pseudogene or an otherwise unique transcript? Is it expressed in NPCs or regulated by DNA methylation?

We have now included a zoom in for Figure 5F with TAF7 labelled and apologise for that oversight. There may indeed be a few smaller peaks over PCDHgamma genes but it's a bit noisy in that region. To the extent that the CUT&RUN assay is quantitative, here MORC2 binding seems to correlate roughly with transcriptional amplitude i.e. TAF7 is much more highly expressed than the PCDHgamma genes, and the peak in TAF7 is far more clear. (See also Reviewer 1 point 8). Regarding how MORC2 is recruited to this locus, we have some ideas but they are speculative at this stage so we're not confident in anything much more than 'long repetitive exon'. There are some old, short L1 fragments at the locus but nothing stands out. (We note that in the mouse genome there are more L1s embedded in the clustered PCDH locus but this is not the case in humans.)

REVIEWER COMMENTS

Reviewer #1 (Remarks to the Author):

The authors have addressed all of my comments and I thank them for not only doing all of the work I requested but also explaining everything clearly.

Geoff Faulkner (University of Queensland)

Reviewer #2 (Remarks to the Author):

This is a much improved and clearer manuscript and I thank the authors for putting in the extra effort. I have a few points below

Comments

1 Introduction much improved – clearer- a pleasure to read

2 If I understand correctly they do not know how the H3K9me3 deposition on young transcriptionally active L1 genes is established and maintained, but they don't think it is HUSH dependent? Unless I missed it in the text, this was not clearly articulated – and it would seem to be important to emphasise this point. I didn't pick it up and it was helpful to have their explanation. Therefore, could it be inserted somewhere around lines 238 – 242 that the H3K9me3 deposition on young transcriptionally active L1 genes is not deposited by HUSH and it's unclear how it is generated?

3 My final query is about how they envisage the DNA methylation is established over the very young L1s. A plausible model is that upon initial integration these transcriptionally active L1s are HUSH sensitive, that HUSH is recruited and at some point DNMT1 is recruited, resulting in locus methylation. Once the locus is fully methylated and transcriptionally inactive HUSH would be lost – the picture they now see – it would be helpful to know if this is their working model? if not - how does the locus get DNA methylated

Reviewer #3 (Remarks to the Author):

I have no additional concerns and recommend publication

Manuscript: "DNA methylation governs the sensitivity of repeats to restriction by the HUSH-MORC2 corepressor"

Re: Author response to reviewer comments (second round)

We express thanks to all the reviewers for their further consideration of our work.

REVIEWER COMMENTS

Reviewer #1 (Remarks to the Author):

The authors have addressed all of my comments and I thank them for not only doing all of the work I requested but also explaining everything clearly.

Geoff Faulkner (University of Queensland)

Thank you again.

Reviewer #2 (Remarks to the Author):

This is a much improved and clearer manuscript and I thank the authors for putting in the extra effort. I have a few points below

Comments

1 Introduction much improved – clearer- a pleasure to read

2 If I understand correctly they do not know how the H3K9me3 deposition on young transcriptionally active L1 genes is established and maintained, but they don't think it is HUSH dependent? Unless I missed it in the text, this was not clearly articulated – and it would seem to be important to emphasise this point. I didn't pick it up and it was helpful to have their explanation. Therefore, could it be inserted somewhere around lines 238 – 242 that the H3K9me3 deposition on young transcriptionally active L1 genes is not deposited by HUSH and it's unclear how it is generated?

Thank you for the suggestion that will indeed make things clearer. We have now added a sentence as follows at the suggested point in the manuscript: *"At the developmental timepoint captured by this somatic model, H3K9me3 maintenance over silent (DNA-methylated) young FL-L1s does not depend on HUSH-MORC2."*

3 My final query is about how they envisage the DNA methylation is established over the very young L1s. A plausible model is that upon initial integration these transcriptionally active L1s are HUSH sensitive, that HUSH is recruited and at some point DNMT1 is recruited, resulting in locus methylation. Once the locus is fully methylated and transcriptionally inactive HUSH would be lost – the picture they now see – it would be helpful to know if this is their working model? if not - how does the locus get DNA methylated

This is an interesting question which we think could be divided up into two: (i) how does DNA of a new L1 integrant get methylated? And (ii) how does a young L1 get methylated during developmental programmes? i.e. arriving at 'the picture we now see' (which we postulate is typical of somatic cells).

For the first part, experiments with transgenes are valuable and DNA methylation of an SVA reporter seemed to be dependent on HUSH as the reviewer previously pointed out (Robbez Mason et al 2018). New L1 integrants were also analysed via transgenes in Liu 2018 and Seczynska 2022 but DNA methylation was not analysed over those reporters – neither in control nor HUSH depleted lines. With regards to endogenous copies, we do see from our analysis of polymorphic L1HS elements identified from whole hNPC genome data that HUSH is needed to restrict these upon genome demethylation (Supp. Fig. 7 reproduced below). These all belong to the L1Ta subfamily of human-specific elements. Since they are absent from the reference genome, they probably represent the youngest L1s (though the insertions could still have happened 1000s of years ago).

For the second part, the model proposed by the reviewer is reasonable. One could also imagine involvement of H3K9me3-associated complexes such as the KRAB-ZNF-TRIM28 system or piRNAs. To highlight these points we have now included the following sentence in the Discussion:

'Whether (re-)establishment of DNA methylation at L1s during development depends on HUSH, or another H3K9me3-associated complex such as the TRIM28 system, is not clear.'

Reviewer #3 (Remarks to the Author):

I have no additional concerns and recommend publication

Thank you again.

REVIEWERS' COMMENTS

Reviewer #2 (Remarks to the Author):

I'm grateful to the authors for having addressed my final concerns and would now strongly recommend publication - really interesting and well performed paper